# MICROSCOPE: EFFICIENT DIFFUSION WITH TWO-STAGE DYNAMICS COMPRESSION FOR HIGH-QUALITY TALKING HEAD GENERATION

## ABSTRACT

The talking head generation task synthesizes videos from a single portrait image and audio input, animating the portrait to deliver the speech content. Non-autoregressive (NAR) approaches for talking head generation have demonstrated impressive quality and generation speeds by producing video frames in parallel, thereby overcoming the error accumulation problems inherent in frame-wise autoregressive (AR) methods. However, NAR methods face limited practical applications due to prohibitive VRAM requirements, especially when generating long sequences ($\geq$ 1000 frames) at high resolution ($512 \times 512$). This paper proposes a novel framework that enables high-quality, non-autoregressive talking head generation while significantly reducing computational resource demands for both training and inference. We enhance efficiency through our Microscope Dynamics Compression Framework (MDCF), a two-stage pipeline achieving 768× compression for pixel-level dynamics latent. Additionally, we demonstrate that this two-stage architecture cannot be ideally optimized via standard end-to-end training. We therefore introduce a two-phase cascade training strategy to stably optimize the MDCF while effectively alleviating error accumulation during multi-stage compression. Experimental results demonstrate that our framework can non-autoregressively generate talking head videos with 1600+ frames at $512 \times 512$ on a 16GB GPU, with state-of-the-art quality and inference speed. Our approach represents a significant advancement toward practical, resource-efficient talking head synthesis for real-world applications. The source code in the supplementary material will be publicly available.

## 1 INTRODUCTION

The talking head generation task aims to synthesize a video of a speaker delivering speech content using only a single portrait image and an audio speech segment as inputs. Recently, talking head generation (Tian et al., 2024; 2025; Xu et al., 2024a; Cui et al., 2024; Jiang et al., 2025; Lin et al., 2025a) has garnered significant attention from researchers due to its importance in digital human interaction, virtual reality, and remote conferencing. Previous methods primarily rely on frame-wise autoregressive models, which achieve temporal coherence by recursively generating video frames (Wei et al., 2024; Stypułkowski et al., 2024). However, these methods have several notable drawbacks: 1) error accumulation that leads to degradation in video quality; 2) slow sequential generation; 3) limitations in long-term dependency modeling.

To address these issues, non-autoregressive (NAR) frameworks such as DAWN (Cheng et al., 2025) have been developed. By generating all video frames in parallel, NAR models not only enhance efficiency but also fundamentally avoid the error accumulation inherent in AR methods. In an NAR framework, the generation of every frame is jointly conditioned on the global context in parallel, rather than depending on a potentially erroneous previous frame. However, NAR methods still face challenges in handling the redundancy of high-dimensional motion representations. Specifically, DAWN uses the diffusion model (Nichol & Dhariwal, 2021) to generate flow-based dynamics representation from audio, employing optical flow as the prior to perform affine transformations on reference images in the latent space. While this approach effectively models global motion at the pixel level with promising vividness and generality, it presents two major problems. First, the method

achieves initial representation compression by leveraging the motion consistency of local pixels, but it relies on the assumption of continuity in neighboring pixels' motion (Siarohin et al., 2021; M & Daniel, 2022). This approximation is valid only for low compression ratios, leading to significant redundancy in dynamics modeling, resulting in high VRAM consumption of the diffusion model and restricting the generation to only a few hundred frames of low-resolution video. Secondly, the diffusion model introduces subtle perturbations during the generation of optical flow fields. These perturbations are subsequently amplified by the image decoder, resulting in pronounced video frame jitter and artifacts. Therefore, current NAR methods urgently require improvements in both efficiency and quality. Talking head generation has widespread applications in mobile devices and edge computing scenarios (Diao et al., 2023). Several methods (Tian et al., 2024; Chen et al., 2024; Cui et al., 2024; Jiang et al., 2025; Tan et al., 2024; Wei et al., 2024) have achieved realistic results by leveraging pre-trained Stable Diffusion Rombach et al. (2022) backbones. However, these approaches face significant limitations for mobile deployment due to their substantial parameter counts and high inference costs (Zhen et al., 2025). Therefore, developing algorithms that simultaneously achieve low inference costs and high-quality visual synthesis represents a critical research imperative for advancing this technology toward practical real-world implementation. Talking head generation has widespread applications in mobile devices and edge computing scenarios (Diao et al., 2023). Several methods (Tian et al., 2024; Chen et al., 2024; Cui et al., 2024; Jiang et al., 2025; Tan et al., 2024; Wei et al., 2024) have achieved realistic results by leveraging pre-trained Stable Diffusion Rombach et al. (2022) backbones. However, these approaches face significant limitations for mobile deployment due to their substantial parameter counts and high inference costs (Zhen et al., 2025). Therefore, developing algorithms that simultaneously achieve low inference costs and high-quality visual synthesis represents a critical research imperative for advancing this technology toward practical real-world implementation.

In this paper, we introduce the Microscope Dynamics Compression Framework (MDCF), a framework that directly solves the VRAM bottleneck to unlock true, global NAR processing for long videos. This novel approach addresses the aforementioned challenges by creating an efficient latent space for diffusion models in image-to-video tasks, such as talking head generation. The MDCF utilizes two cascaded sub-compressors, functioning like a microscope's objective lens and eyepiece, achieving multiplicative compression with a ratio of 768. This paves the way for NAR diffusion-based talking head generation architectures to produce long videos with high-resolution content. Furthermore, this approach's capacity for global pixel-level motion modeling offers potential as an efficient acceleration framework for a broader range of image-to-video tasks. The method is based on the following core insights: 1) the motion of neighboring pixels exhibits local continuity, which can be initially modeled by slightly downsampled optical flow to capture pixel-level dynamics patterns; 2) the redundant information in high-dimensional pixel-level dynamics features can be further compressed through neural network latent space mapping. Based on two key assumptions, the cascaded compression approach achieves an overall compression ratio that exceeds the limitations of individual sub-compressors while preserving high decompression quality.

To achieve efficient training of MDCF and avoid cumulative errors in the multi-level compression process, we formulated a specialized training paradigm, namely the Two-Phase Cascaded (TPC) training strategy. The core idea of TPC is to train each sub-compressor separately while maintaining consistent image-level supervision, thereby stabilizing model training and promoting inter-module cooperation. Incorporating the proposed MDCF, our overall framework achieves the generation of over 1600 frames of $256 \times 256$ resolution video on a V100 16G GPU, with memory usage reduced by 4.69 times and inference speed increased by 3.12 times compared to the baseline. The core contributions can be summarized as follows:

- Introduced the Microscope Dynamics Compression Framework (MDCF), which achieves pixel-level dynamics modeling with a high-quality $768\times$ compression ratio.
- Proposed the Two-Phase Cascaded (TPC) Training Strategy for MDCF, which mitigates error accumulation while reducing training costs and improving stability.
- Significantly reduced VRAM requirements for generating high-resolution, long-duration talking head videos, while achieving quality that matches or exceeds the state-of-the-art (SOTA).

## 2 RELATED WORKS

**Talking Head Generation.** Early approaches relied on GANs or neural rendering (Zhou et al., 2020; Guo et al., 2021), while diffusion models have recently become dominant for their superior visual quality (Tian et al., 2024; Lin et al., 2025b; Ma et al., 2024). Diffusion-based talking head generation methods can be categorized as autoregressive (AR) or non-autoregressive (NAR) (Cheng et al., 2025). AR methods Tian et al. (2024); Xu et al. (2024b); Stypułkowski et al. (2024) generate video frames sequentially and concatenate them to form the complete video. In contrast, NAR methods (Cheng et al., 2025; Du et al., 2023) generate video frames in parallel. NAR approaches offer three key advantages: they reduce error accumulation during generation, utilize contextual information more effectively, and maximize hardware computational efficiency through parallel processing. Despite maintaining quality in long video generation, NAR methods require simultaneous processing of all frames, resulting in excessive memory consumption that prohibits the generation of longer content. Research indicates that reducing the dimensionality of diffusion model outputs can substantially lower computational costs (Rombach et al., 2022). Some studies leverage facial prior features or decouple appearance and motion information to obtain low-dimensional motion representations, thereby simplifying the generation process (Xu et al., 2024b; Liu et al., 2024b; Ma et al., 2023). However, approaches based on facial priors often compromise model generalization ability, particularly when processing non-centered facial scenes (Liu et al., 2024b; Ma et al., 2023). This work focuses on developing efficient latent space representations at the pixel level, and thus addresses the memory consumption of NAR methods during long video inference while enhancing the generation quality.

**Diffusion Model with Efficient Latent.** The field of diffusion models in visual generation is experiencing rapid evolution, with continuous efforts to enhance model performance and efficiency. Stable Diffusion (Rombach et al., 2022) pioneered the use of VAEs to compress content in latent space, substantially improving generation quality while accelerating inference speed. However, the compression algorithm with VAE typically employs an $8 \times 8$ spatial compression, as higher compression ratios significantly degrade image reconstruction quality (Chen et al., 2025). Further research (Chen et al., 2025; Xie et al., 2025) indicates that designing auto-encoders with high compression ratios is essential for generating high-resolution content. This has led to the development of efficient auto-encoders with downsampling factors greater than $32 \times 32$, while shifting the information burden to the channel dimension. Consequently, the dimensionality of the latent states is not completely compressed in terms of the number of tokens. For spatiotemporal information, approaches like, Wan (Wang et al., 2025), and Step-Video-T2V (Ma et al., 2025) compress temporal and spatial information simultaneously through 3D-VAE architectures. Despite significant progress in image and video generation, researchers have not adequately explored efficient compression techniques for global motion information in image-to-video applications, especially for talking head generation. This paper addresses this gap by proposing a novel two-stage compression strategy based on dual perspectives. Our approach achieves a remarkable compression ratio of $768\times$ while focusing on spatial information compression, significantly enhancing the generation efficiency of diffusion models for talking head applications.

## 3 METHOD

In this section, we present our method in three parts: (1) we introduce the Microscope Dynamics Compression Framework (MDCF); (2) we detail our Two-Phase Cascaded (TPC) training strategy for MDCF; and (3) we outline our overall pipeline for talking head generation.

### 3.1 MICROSCOPE DYNAMICS COMPRESSION FRAMEWORK

In this section, we propose the Microscope Dynamics Compression Framework (MDCF) to construct an efficient dynamics latent space for audio-driven talking head generation tasks. This two-stage approach is critical, as applying a standard single-stage compressors directly results in significant quality degradation at high compression ratios (Table 5). As shown in Figure 1, this architecture consists of two cascaded compression stages: the flow-aware dynamics extractor (FDE) and the latent motion auto-encoder (LMAE). This design enables the diffusion model to operate in a highly compressed space, thereby substantially improving generation efficiency.

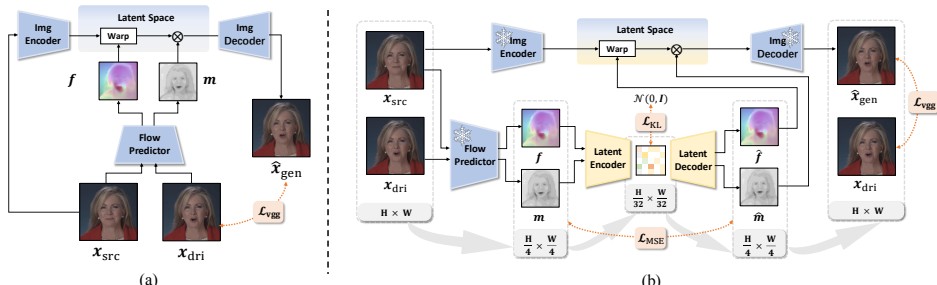

Figure 1: (a) Pipeline of the Flow-aware Dynamics Extractor (FDE). (b) Pipeline of the Microscope Dynamics Compression Framework (MDCF). We first train the Flow-aware Dynamics Extractor (FDE) in (a) to capture essential motion patterns. Then the FDE is frozen while optimizing the Latent Motion Auto-Encoder (LMAE) in (b).

**Stage 1: Flow-aware Dynamics Extractor.** The first stage utilizes the Flow-aware Dynamics Extractor (FDE) to develop a pixel-level dynamics representation, illustrated in Figure 1 (a). This representation captures the motion patterns in a talking head video relative to a reference portrait. Notably, this dynamics representation enables the reconstruction of the entire video sequence from a single corresponding portrait. The FDE's core affine transformation (warping) operation $\mathcal{A}$ does not occur in the raw pixel space, but rather in a downsampled latent space, as illustrated in Figure 3.3 (a) The Image Encoder $\mathcal{E}$ first maps the source image $x_{src}$ into this latent space. This encoding process, typically involving convolutional downsampling, inherently aggregates information from neighboring pixels into latent features. This stage utilizes a low spatial downsampling rate (e.g., 4x) for both the latent image features and the corresponding dynamics representations. This is a critical design choice: a high compression rate (heavy downsampling) at this stage would create a low-resolution latent space, severely degrading the precision of the warping operation $\mathcal{A}$ and failing to capture fine-grained motion details, such as erroneous lip movements. The extracted information primarily includes the optical flow field $\boldsymbol{f} \in \mathbb{R}^{N \times H' \times W' \times 2}$ and the occlusion map $\boldsymbol{m} \in \mathbb{R}^{N \times H' \times W' \times 1}$, which are used respectively for warping the original image and repairing occluded areas, in accordance with the definition in Siarohin et al. (2021). Crucially, the inclusion of the occlusion map $\boldsymbol{m}$ allows the model to handle complex non-flow dynamics, such as repairing newly exposed regions (dis-occlusions) during mouth or eye movements, which is a known limitation of pure flow-only methods. Its mathematical expression is given by:

$$\boldsymbol{f}, \boldsymbol{m} = \mathcal{P}(\boldsymbol{x}_{\mathrm{src}}, \boldsymbol{x}_{\mathrm{dri}}), \tag{1}$$

$$\hat{\boldsymbol{x}}_{\mathrm{gen}} = \mathcal{D}\Big(\mathcal{A}\big(\mathcal{E}(\boldsymbol{x}_{\mathrm{src}}), \boldsymbol{f}\big) \otimes \boldsymbol{m}\Big), \tag{2}$$

where $\mathcal{P}$ and $\mathcal{D}$ denote the flow predictor and image decoder of FDE, the $\mathcal{A}$ is the affine transformation operation, "$\otimes$" is the element-wise production, $\boldsymbol{x}_{\mathrm{src}}$ and $\boldsymbol{x}_{\mathrm{dri}}$ represent the source and driving images, respectively, and $\hat{\boldsymbol{x}}_{\mathrm{gen}}$ is the generated image.

**Stage 2: Latent Motion Auto-Encoder.** After extracting dynamics representations and achieving initial compression through the FDE, we introduce the Latent Motion Auto-Encoder (LMAE) to further compress the dynamics representations and reduce spatial redundancy. Specifically, the LMAE is based on the variational auto-encoder (VAE) (Kingma & Welling, 2022) architecture. It consists of a latent encoder (LE) and a latent decoder (LD), which are responsible for compressing and decompressing the motion representation. The compression and decompression process can be summarized as:

$$\hat{\boldsymbol{z}} = \mathcal{E}_{\mathrm{LE}}([\boldsymbol{f}; \boldsymbol{m}]), \qquad [\hat{\boldsymbol{f}}; \hat{\boldsymbol{m}}] = \mathcal{D}_{\mathrm{LD}}(\hat{\boldsymbol{z}}), \tag{3}$$

where $\hat{\boldsymbol{z}}$ is the dynamics latent of LMAE, "$[\ ;\ ]$" refers to the concatenation operation, the $\mathcal{E}_{\mathrm{LE}}$ and $\mathcal{D}_{\mathrm{LD}}$ are the encoder and decoder of LMAE. We observed that when using diffusion to directly generate representations constructed by FDE, the resulting videos exhibit noticeable visual jitters. We attribute this to the fact that the minor random perturbations in temporal modeling by the diffusion model are amplified by the image decoder of the FDE. Therefore, we fully exploit the sampling mechanism of the VAE structure, and turn the LMAE into a natural low-pass filter to enhance video

quality. Specifically, the dynamics latent of LMAE is sampled by:

$$\hat{z} = \mu([\,\boldsymbol{f}\,;\,\boldsymbol{m}\,]) + \sigma([\,\boldsymbol{f}\,;\,\boldsymbol{m}\,]) \cdot \epsilon, \quad \epsilon \sim \mathcal{N}(0, I), \tag{4}$$

where $\mu$ and $\sigma$ are the mean and variance encoders of $\mathcal{E}_{\mathrm{LE}}$. During training, the input of LD inherently contains random noise, which means the LD has the potential to suppress the noise from the input. Building on the aforementioned observation, we conduct the generation process within the latent space of $\hat{z}$ and utilize the decoder as a low-pass filter. This approach effectively suppresses the high-frequency temporal noise introduced by the diffusion model.

The overall architecture is analogous to the stacked lenses of a microscope. In this design, integrating the FDE and LMAE multiplies their individual compression ratios, thereby enhancing the system's overall compression capability, similar to how multiple lenses increase magnification in a microscope. Consequently, the MDCF achieves a high degree of compression, providing a solid foundation for subsequent generative tasks. Specifically, the overall compression ratio can be expressed as: $C_{\mathrm{MDCF}} = C_{\mathrm{FDE}} \times C_{\mathrm{LMAE}}$, where $C_{\mathrm{MDCF}}$, $C_{\mathrm{FDE}}$, $C_{\mathrm{LMAE}}$ are compression factors of MDCF, FDE and LMAE.

## 3.2 Two-Phase Cascaded Training

Given that the two stages of MDCF utilize distinct principles for compressing motion features, we have observed that training MDCF directly in an end-to-end manner poses convergence challenges. Additionally, integrating both stages into a single end-to-end training process significantly increases computational costs. To achieve more efficient and stable optimization, we propose the Two-Phase Cascaded (TPC) training approach. Overall, MDCF adopts an unsupervised training mode with a process divided into two phases.

**Phase One: FDE Training.** In the first phase, we train the FDE independently. The corresponding loss function employs a perceptual loss:

$$\mathcal{L}_{\mathrm{FDE}} = \mathcal{L}_{\mathrm{vgg}}\Big(\mathcal{D}\big(\mathcal{A}\big(\mathcal{E}(\boldsymbol{x}_{\mathrm{src}}), \boldsymbol{f}\big) \otimes \boldsymbol{m}\big), \boldsymbol{x}_{\mathrm{dri}}\Big), \tag{5}$$

where $\mathcal{L}_{\mathrm{vgg}}$ represents the perceptual loss calculated based on the VGG network (Johnson et al., 2016).

**Phase Two: LMAE Training.** In the second phase, we freeze the pre-trained FDE from the first phase and incorporate LMAE training, as shown in Figure 1 (b). During training, we use dynamics representations calculated online by FDE to train LMAE for reconstruction. After training, LMAE is expected to fulfill three primary functions: (1) reconstructing the flow-based dynamics extracted by FDE with minimal loss; (2) working synergistically with FDE to recover the target image with minimal loss; and (3) maintaining robustness against potential noise perturbations in the latent space.

First, for latent reconstruction, we apply a Mean Squared Error (MSE) based reconstruction loss, defined as:

$$\mathcal{L}_{\mathrm{MSE}} = \big\| \mathcal{D}_{\mathrm{LD}}(\hat{z}) - [\,\boldsymbol{f}\,;\boldsymbol{m}\,] \big\|_2^2 \tag{6}$$

However, direct reconstruction of dynamics may lead to error accumulation within multi-stage compression. To address this issue, we propose the Image Guided Consistency (IGC) loss. This approach enables the LMAE and FDE to share a consistent training objective: reconstructing the driving image with minimal loss. Since LMAE does not directly process image data, we utilize the pre-trained FDE to propagate the gradient from image-level supervision to LMAE. The IGC loss is defined as:

$$\mathcal{L}_{\mathrm{IGC}} = \mathcal{L}_{\mathrm{vgg}}\Big(\mathcal{D}\big(\mathcal{A}\big(\mathcal{E}(\boldsymbol{x}_{\mathrm{src}}), \hat{\boldsymbol{f}}\big) \otimes \hat{\boldsymbol{m}}\big), \boldsymbol{x}_{\mathrm{dri}}\Big), \tag{7}$$

where $\hat{\boldsymbol{f}}, \hat{\boldsymbol{m}}$ are calculated by Equation 3. The introduction of the IGC loss enables LMAE to directly leverage image-level supervision while promoting more effective cooperation with FDE, ultimately enhancing the system's reconstruction quality. Additionally, to prepare LMAE for decoding slightly perturbed inputs during inference, we implement a KL divergence regularization loss on LMAE's latent variables. This prevents the latent encoder from producing extremely low variance during training, which would otherwise compromise the decoder's robustness to noise. The regularization term is defined as:

$$\mathcal{L}_{\mathrm{KL}} = D_{\mathrm{KL}}\Big(q(\hat{z} \,|\, [\,\boldsymbol{f}\,;\boldsymbol{m}\,]) \,\big\|\, p(\hat{z})\Big), \quad p(\hat{z}) = \mathcal{N}(0, \boldsymbol{I}), \tag{8}$$

where $p, q$ are the probability density functions. Finally, the total LMAE loss can be expressed as:

$$\mathcal{L}_{\text{total}} = \lambda_{\text{KL}}\mathcal{L}_{\text{KL}} + \lambda_{\text{MSE}}\mathcal{L}_{\text{MSE}} + \lambda_{\text{IGC}}\mathcal{L}_{\text{IGC}}, \tag{9}$$

where $\lambda_{\text{KL}}$, $\lambda_{\text{MSE}}$, $\lambda_{\text{IGC}}$ are hyperparameters of loss weight.

In summary, through Two-Phase Cascaded (TPC) training, we effectively mitigate the convergence difficulties and computational overhead associated with the direct end-to-end training of MDCF, while ensuring high-quality performance in reconstruction tasks.

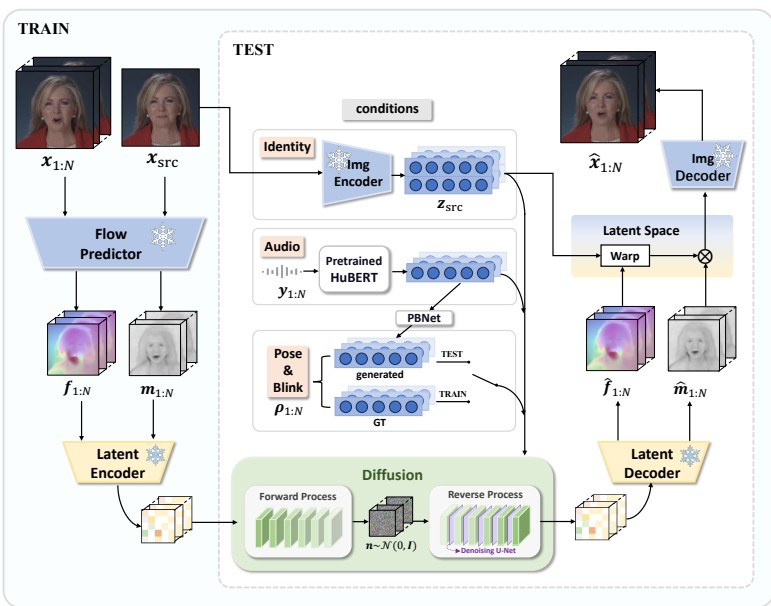

Figure 2: Our proposed method follows a three-step pipeline. (1) Compression: We use the encoder component of MDCF to perform two-stage compression, generating highly compressed dynamics latent. (2) Generation: A diffusion model generates the corresponding dynamics latent using the following inputs: source image, audio, head pose, and blink signal. (3) Decompression: We employ the decoder component of MDCF for two-stage decompression, reconstructing the target video.

## 3.3 Efficient Latent Diffusion for Audio-Driven Talking Head Generation

By introducing MDCF and employing an appropriate training strategy, we have successfully developed a highly compressed dynamics representation. This representation serves as the generation target for the Diffusion Model (DM), enabling training within a compact latent space. This method significantly reduces computational costs for both training and inference. Ultimately, we use MDCF to decompress the DM's output, thereby reconstructing the complete talking head video. The overall pipeline is shown in Figure 2. The generation process of the DM can be expressed as:

$$\hat{z}_{1:N} = \text{DM}\big(\mathcal{E}(\boldsymbol{x}_{\text{src}}), \boldsymbol{y}_{1:N}, \boldsymbol{\rho}_{1:N}\big), \tag{10}$$

where $N$ represents the number of video frames; DM represents the diffusion model; $\boldsymbol{y}$ represents the embedding of audio signals, while $\boldsymbol{\rho}$ corresponds to head pose and eye-blinking control signals.

During the training process, we initially load the video clip and extract its latent states online to use as labels by the frozen MDCF. For supervision, we employ the standard diffusion loss as outlined in DDPM (Nichol & Dhariwal, 2021). The diffusion model takes audio, the first video frame, head motion, and eye-blinking signals as conditions to denoise the latent space sequence $\hat{z}$. We extract pose and eye-blinking signals from real video to guide the diffusion model, enabling precise control over target poses and blinking patterns.

During the inference phase, we use diffusion model to generate the dynamics latent. Subsequently, the MDCF module conducts a two-stage decompression process on the motion representations.

First, LMAE performs initial decompression to recover flow-based dynamics representations. Then, the FDE module utilizes these representations to warp the source image embedding and repair occlusion regions via the image decoder, ultimately reconstructing the complete talking head video. Additionally, we use the pose and blink generation network (PBNet) (Cheng et al., 2025) to synthesize head poses and eye-blinking action sequences from audio, driving the diffusion model to generate natural and coherent eye-blinking and head poses.

## 4 EXPERIMENT

### 4.1 IMPLEMENTATION

We train our method on the HDTF dataset (Zhang et al., 2021), and randomly divide it into training and testing sets with a 9:1 ratio. We use the HuBERT (Hsu et al., 2021) to embed the audio signal before training. The MDCF module operates in two stages: 1) First stage: We extract dynamics features using a $4\times$ downsampling rate, creating an initial motion representation $[\boldsymbol{f}; \boldsymbol{m}]$ of size $\frac{H}{4} \times \frac{W}{4} \times 3$. The first two channels encode the optical flow field, while the third channel represents the occlusion map. 2) Second stage: We compress the dynamics features with a further $8\times$ downsampling rate, yielding the latent $\hat{\boldsymbol{z}}$ of size $\frac{H}{32} \times \frac{W}{32} \times 4$, achieving the compression ratio of 768 $(32 \times 32 \times \frac{3}{4})$. The loss function integrates multiple metrics with the following weight distribution: $\lambda_{\text{KL}} = 1 \times 10^{-4}$, $\lambda_{\text{MSE}} = 1$, $\lambda_{\text{IGC}} = 1$. These weights were chosen to balance the model's objectives rather than requiring exhaustive tuning. $\lambda_{KL} = 1 \times 10^{-4}$ is a standard, small weight used in VAE training to provide light regularization against posterior collapse without harming reconstruction. $\lambda_{MSE}$ and $\lambda_{IGC}$ were set to 1.0 as both feature-level reconstruction ($\mathcal{L}_{MSE}$) and final image-level consistency ($\mathcal{L}_{IGC}$) are considered equally critical. For the diffusion model, we employ VDM (Ho et al., 2022) with a 3D-Unet backbone to generate the highly compressed latent provided by MDCF. We use the same training strategy as DAWN (Cheng et al., 2025) to train the diffusion model for fair comparison. The entire training process was conducted using four V100 32G GPUs. Additionally, to evaluate cross-dataset generalization, we test our model (trained only on HDTF) on the VoxCeleb2 Chung et al. (2018) dataset. We randomly select 400 videos in VoxCeleb2 for evaluation.

To evaluate the overall quality of images and videos, we employ the FID (Heusel et al., 2017) and FVD (Unterthiner et al., 2019) metrics, respectively. The FVD is assessed at different scales, specifically 16 frames (FVD-16) and 32 frames (FVD-32). To evaluate the synchronization between the audio and lips, we utilize the sync-net (Chung & Zisserman, 2017) to calculate the synchronization confidence score $\text{LSE}_{\text{C}}$ and distance score $\text{LSE}_{\text{D}}$. To ensure fair comparison across different methods, all videos are uniformly resized to $256 \times 256$ during testing before evaluation. All inference experiments were conducted using a V100 16GB GPU. Notably, all comparative methods generate videos of 1600 frames at 25 fps, with the exception of DAWN, which is constrained by VRAM limitations and cannot produce videos exceeding 200 frames at $256 \times 256$ resolution.

### 4.2 OVERALL COMPARISON

**Comparison with the SOTAs.** In this section, we compare our method at resolution of $256 \times 256$ and $512 \times 512$ with several state-of-the-art (SOTA) methods: Audio2Head (Wang et al., 2021), Sad-Talker (Zhang et al., 2023), Hallo (Xu et al., 2024a), Hallo2 (Cui et al., 2024), EchoMimic (Chen et al., 2024), AniTalker (Liu et al., 2024b), and DAWN (Cheng et al., 2025). Notably, both Hallo, Hallo2 and EchoMimic incorporate the pre-trained stable diffusion model, inheriting strong visual generation capabilities. As shown in Table 1, our proposed method quantitatively outperforms existing techniques in both image and video quality metrics, while maintaining comparable accuracy in lip movement. Furthermore, we conducted a qualitative comparison with several recent approaches, as illustrated in Figure 3. The results indicate that our method achieves realistic portrait animation without the need for input image cropping, highlighting its distinct advantages over current techniques. We also performed a computational efficiency comparison with DAWN, a state-of-the-art non-autoregressive method. As presented in Table 4, for the task of generating a 200-frame video at $256 \times 256$ resolution, our method achieves: 1) 4.69-fold reduction in VRAM consumption compared to DAWN's 18.69GB requirement. 2) 3.12-times speedup in inference time. These findings underscore the superior efficiency of our approach, making it more accessible for practical applications. Figure 4a presents a comprehensive speed comparison of previous SOTA

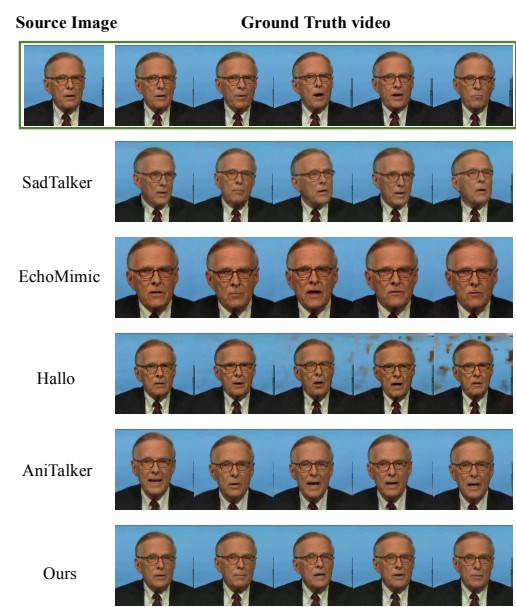

**Source Image**  **Ground Truth video**

SadTalker

EchoMimic

Hallo

AniTalker

Ours

Figure 3: Qualitative comparison with previous SOTAs.

Table 1: Quantitative comparison on HDTF.

| Method | FID↓ | FVD$_{16}$↓ | FVD$_{32}$↓ | LSE$_C$↑ | LSE$_D$↓ | CSIM↑ |
|---|---|---|---|---|---|---|
| Audio2Head | 30.10 | 122.26 | 205.42 | 6.88 | **7.58** | 0.705 |
| SadTalker | 26.11 | 97.43 | 187.43 | 6.27 | 8.03 | 0.767 |
| Hallo | 41.32 | 154.38 | 217.6 | **7.49** | 7.94 | 0.709 |
| EchoMimic | 32.80 | 139.00 | 178.16 | 6.69 | 8.27 | 0.731 |
| AniTalker | 44.42 | 133.36 | 208.36 | 6.04 | 9.27 | 0.725 |
| DAWN* | 11.80 | 68.07 | 105.20 | 7.20 | 7.80 | 0.790 |
| Hallo2* | 19.10 | 113.30 | 164.58 | 7.40 | 7.697 | 0.789 |
| Ours-256 | **11.22** | **44.60** | **60.28** | 7.33 | 7.85 | 0.791 |
| Ours-512 | 12.57 | 51.02 | 65.40 | 7.14 | 7.95 | **0.806** |

Table 2: Comparison of compression methods with different ratios.

| Method | FID↓ | FVD$_{16}$↓ | FVD$_{32}$↓ | LSE$_C$↑ | LSE$_D$↓ |
|---|---|---|---|---|---|
| GT | 0 | 0 | 0 | 8.30 | 7.05 |
| FDE (d/4) | 7.90 | 21.18 | 30.45 | 7.92 | 7.89 |
| FDE (d/8) | 10.33 | 35.07 | 62.60 | 6.89 | 8.26 |
| VAE (d/32) | 24.26 | 62.84 | 100.61 | 7.64 | 7.52 |
| MDCF (d/16) | 7.85 | 21.31 | 30.40 | **7.95** | 7.38 |
| MDCF (d/32) | **7.84** | **20.84** | **29.46** | 7.92 | **7.36** |

methods, all generating 200-frame (8-second) videos. Our framework demonstrates the fastest inference time among open-source diffusion-based talking head generation approaches, being second only to Audio2Head, a GAN-based method. This speed difference is expected, as GAN-based methods are single-pass, whereas diffusion models require iterative denoising. Our objective was not to outperform GANs on speed, but to address the critical inefficiency of diffusion-based SOTA quality methods. Furthermore, methods like Audio2Head utilize flow-based representations with low compression ratios. While feasible for GANs, this non-compact latent space creates a severe bottleneck for diffusion models, which must iteratively denoise this large representation, leading to excessive VRAM usage (as seen in DAWN in Table 4) and difficulty scaling to high-resolution training. Our MDCF directly solves this bottleneck. We also provide extensive comparisons to other LDM-based methods, such as Hallo and EchoMimic (see Table 1 and Figure 3), demonstrating superior quality and efficiency. Moreover, our latent space compression technology significantly reduces training resource requirements. This enables stable training at resolutions of $512 \times 512$ with limited computational resources while maintaining excellent video generation performance. These experimental results clearly demonstrate the advantages of our method: superior generation quality and enhanced resource efficiency.

To evaluate the generalization capacity of our framework, we conducted a zero-shot cross-dataset evaluation on VoxCeleb2. The model, trained exclusively on HDTF, was tested on the VoxCeleb2 test set without any finetuning. We compare against recent SOTA methods Hallo Xu et al. (2024a) and Hallo2 Cui et al. (2024). As shown in Table 3, our method demonstrates strong generalization to unseen identities and environments. The results show that even without training on VoxCeleb2, our method achieves comparable or superior performance across all metrics, including significantly better FID/FVD scores and robust identity similarity (CSIM). This confirms that our MDCF framework learns a generalizable representation of motion dynamics.

Table 3: Cross-dataset evaluation on Voxceleb2 with previous SOTAs.

| Method | FID↓ | FVD$_{16}$↓ | FVD$_{32}$↓ | LSE$_C$↑ | LSE$_D$↓ | CSIM↑ |
|---|---|---|---|---|---|---|
| Hallo | 34.59 | 323.65 | 545.94 | 4.72 | 9.23 | 0.610 |
| Hallo2 | 24.62 | 233.45 | 389.77 | 4.93 | 9.50 | 0.643 |
| ours | 18.54 | 207.83 | 288.04 | 4.80 | 9.39 | 0.662 |

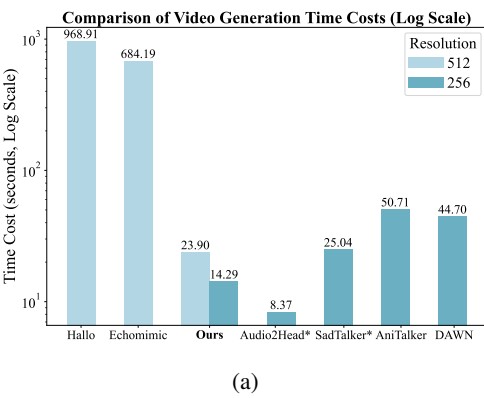
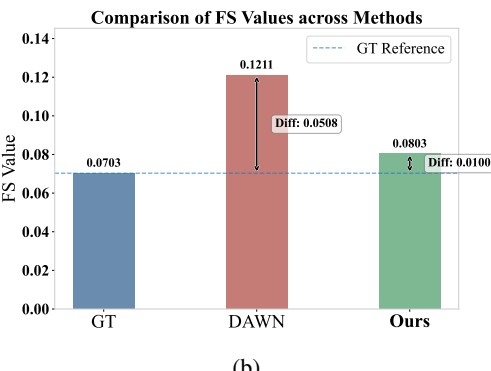

(a)                                                                    (b)

Figure 4: (a) Inference speed comparison between our method and previous state-of-the-art approaches for generating 8-second videos. The symbol "*" represents the GAN-based method, while the others are diffusion-based. (b) Comparison of Flow Smoothness (FS) metrics.

Table 4: Computational efficiency comparison between our method and DAWN for 8-second video generation. MEM indicates peak GPU VRAM consumption during inference.

Table 5: Quantitative study on HDTF dataset of compression stages and training strategies for MDCF.

|  | Resolution | Length | MEM | Time |
|---|---|---|---|---|
| DAWN | 128 | 200 | 5.43G | 8.3s |
|  | 256 | 200 | 18.69G | 44.7s |
| Ours | 256 | 200 | 3.98G | 14.3s |
|  | 256 | 1600 | 4.57G | 90.6s |
|  | 512 | 200 | 10.00G | 23.9s |
|  | 512 | 1600 | 14.50G | 153.0s |

| Method | FID↓ | FVD$_{16}$↓ | FVD$_{32}$↓ | LSE$_C$↑ | LSE$_D$↓ |
|---|---|---|---|---|---|
| GT | 0 | 0 | 0 | 8.30 | 7.05 |
| LIA | 26.64 | 95.75 | 178.62 | 6.69 | 8.12 |
| FDE | 7.90 | 21.18 | 30.45 | 7.92 | 7.89 |
| MDCF | **7.84** | **20.84** | **29.46** | 7.92 | **7.36** |
| w/o IGC loss | 8.79 | 27.19 | 40.69 | 7.58 | 7.59 |
| w/o TPC | 12.75 | 42.00 | 64.52 | 7.00 | 7.59 |

**Comparison of Flow Smoothness.** This section aims to analyze video jitter and artifacts caused by randomness during the generation process of diffusion models in generated videos. Such jitter is often not prominently reflected in visually dominant metrics such as FID, as these metrics primarily focus on the similarity of perceptual features and fail to capture subtle motion discontinuities. We have provided the visualization of this problem in Appendix A.2. We observe that video jitter exhibits distinct features in the optical flow field: when jitter occurs, numerous small spikes appear in the optical flow field, causing naturally smooth motion to appear rough. Based on this phenomenon, we specifically designed an evaluation metric, Flow Smoothness (FS), to detect and quantify this issue, with detailed definitions in Appendix A.3. The core principle of this metric is that by examining the spatial gradient magnitude of the optical flow field, it can effectively indicate the smoothness of motion. Because of the inherent motion present in talking head videos, which naturally results in a basic optical flow gradient, it is ideal for the FS value of the generated videos to closely match the FS value of the ground truth videos. We used the FS metric to evaluate the generated video from DAWN and our method, as shown in Figure 4b. According to the results, it can be observed that the videos generated by our method are closer to Ground Truth (GT) videos in terms of flow smoothness. To empirically validate the proposed FS metric, we performed a user study in which participants provided perceptual judgments on video stability (denoted as "V-stab" in Table 6). The results show that human evaluations exhibit a strong positive correlation with our FS metric, thereby confirming its efficacy in quantifying perceptually relevant motion stability. This further validates the effectiveness of our adopted MDCF strategy, which implicitly achieves low-pass filtering through the sampling mechanism during latent motion decoder training.

### 4.3 ABLATION STUDY

In this section, we conduct a series of ablation studies to validate our design choices.

**Motivation for Multi-Stage Design.** Our primary motivation stems from the inherent limitations of single-stage compression. As shown in Table 2, single-stage methods suffer substantial quality degradation at high compression ratios. For instance, a single-stage FDE at a $64\times$ ratio ("FDE(d/8)") fails because its core assumption of local motion consistency breaks down. This result also empirically validates our choice of a slight $4\times$ downsampling ($\frac{H}{4} \times \frac{W}{4}$) for the FDE stage, as a higher rate (like 8x) already compromises motion fidelity. Similarly, a single-stage VAE at a $768\times$ ratio ("VAE(d/32)") shows a collapse in performance across all metrics, rendering it unsuitable for subsequent generation tasks. This demonstrates a fundamental trade-off: high compression ratios in a single step inevitably lead to unacceptable information loss. Our multi-stage design can effectively overcome such a bottleneck.

**Effectiveness of MDCF**. Our experiments validate that the proposed MDCF successfully mitigates this issue. The results show that MDCF maintains high reconstruction quality throughout its cascaded compression stages, avoiding the significant degradation seen in single-stage approaches. Remarkably, its final video and image quality even slightly surpass that of a single-stage FDE at a much lower compression ratio. We attribute this to the Image-Guided Consistency (IGC) loss, which guides the LMAE to effectively compensate for information loss from the FDE stage. We also compare our reconstruction-focused approach against state-of-the-art video-driven talking head methods like LIA (Wang et al., 2024), which often rely on complex appearance-motion disentanglement. Our method significantly outperforms LIA. This advantage arises because MDCF bypasses the error-prone disentanglement process—a common performance bottleneck—by directly optimizing for video reconstruction. This streamlined strategy enhances the overall quality and robustness of the system. Finally, we ablate our key components. Removing the IGC loss ("w/o IGC loss") leads to a noticeable increase in compression loss, confirming its crucial role in stabilizing multi-stage performance. Furthermore, replacing our Two-Phase Cascaded (TPC) training strategy with a standard end-to-end approach ("w/o TPC") results in poor convergence. This underscores the efficacy of TPC in optimizing our complex, multi-stage architecture. To further analyze the compression trade-off, we additionally tested MDCF at a $192\times$ ($4 \times 4$ for FDE and $4 \times 4 \times \frac{3}{4}$ for LMAE) compression ratio (see Table 2). We observed that the performance at $192\times$ is very close to that of our 768x compression setting. This indicates that our method provides robust, high-quality reconstruction across a wide range of high compression ratios, rather than exhibiting a steep trade-off. This strongly validates the design of our multi-stage framework.

## 5 CONCLUSION

This paper introduces an efficient framework for talking head avatar generation that produces high-resolution, high-quality long videos while maintaining low computational costs and fast inference speeds. Our approach consists of two key components: a Microscope Dynamics Compression Framework (MDCF) and a video diffusion model. The MDCF module hierarchically compresses global motion representations from two distinct perspectives, achieving an impressive compression ratio of up to 768:1. To stably optimize this multi-stage architecture, which we show fails to converge with standard end-to-end training , we introduce the necessary methodological contribution of a Two-Phase Cascaded (TPC) training strategy. The diffusion model operates within the compact motion latent space created by MDCF, with video generation achieved through MDCF's two-level decompression process. Experimental results show that our framework achieves or surpasses state-of-the-art performance in generation quality while enabling fast inference. Beyond talking head avatar generation, this approach shows promise for various image-to-video tasks, potentially accelerating the adoption of these technologies in resource-constrained environments, such as mobile devices and edge computing scenarios.

## 6 REPRODUCIBILITY STATEMENT

To ensure the reproducibility of the results in this work, all necessary resources have been organized in the supplementary materials and will be made publicly available upon paper acceptance. The supplementary materials include complete source code for implementing all proposed models, experimental pipelines, and detailed hyperparameter settings. Upon acceptance, we will open-source the aforementioned code, hyperparameter documentation as well as pretrained model via a public code repository, with the repository link to be provided promptly.

## 7 ETHICS STATEMENT

We acknowledge the ethical considerations associated with talking head generation technology.

**Data Transparency.** The datasets used in this work, HDTF and VoxCeleb2 (used for generalization testing), are publicly available benchmarks intended for academic research.

**Potential Misuse.** Like all talking head synthesis techniques, our method has the potential for misuse in creating "deepfakes" or misleading content. We recognize the importance of addressing these risks.

**Positive Applications.** We believe the primary applications of this technology are beneficial. Our work aims to advance positive use cases, such as creating realistic digital humans for virtual assistants, enhancing accessibility tools for communication, improving virtual reality conferencing, and providing engaging educational content.

**Responsible Research.** We are committed to responsible research practices. By focusing on computational efficiency, our work also aims to democratize access to this technology for researchers in resource-constrained environments. We support future work in developing robust watermarking and detection mechanisms to mitigate the risks of malicious use.

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

# A APPENDIX

## A.1 ANALOGY OF "MICROSCOPE": THE LIMITATION OF SINGLE-STAGE COMPRESSION

The reason why microscopes do not use a single lens for "magnification" can be explained as follows: (1) Single lenses exhibit severe spherical aberration and chromatic aberration under high magnification, causing degraded image quality; (2) Numerical aperture, which quantifies light-gathering capability, faces inherent physical constraints in single lenses due to geometric and material limitations, while multi-lens systems can achieve higher effective values through optimized design to meet resolution requirements. These issues reflect fundamental physical limitations of single-stage systems. Modern optical microscopes utilize multi-stage magnification, enabling optimal performance of each component. Thus, multi-lens systems demonstrate clear advantages over single-lens configurations for magnification applications.

Analogously, in our video representation compression task, single-stage methods (e.g., flow-based and VAEs) show marked performance degradation at high compression ratios. As summarized in Table 2. The single-stage FDE or VAE compressor shows marked performance degradation at high compression ratios. The results underline the same principle as the microscope analogy: single-stage designs run into intrinsic limits, whereas multi-stage compression maintains quality at higher ratios and thus better supports subsequent talking-head generation.

## A.2 THE QUALITATIVE STUDY WITH DAWN

In this section, we conduct a qualitative study comparing our method with DAWN regarding motion perturbations and jitters, as shown in Figure 5. From the results, we observe that these issues cannot be detected merely from the visual perspective (left half of Figure 5). However, these problems are quite evident at the motion level. In the optical flow visualization (right half of Figure 5), DAWN clearly exhibits more motion artifacts and jitters, which is also reflected in the supplementary video. In contrast, our proposed method produces smoother motion, demonstrating closer resemblance to the ground truth.

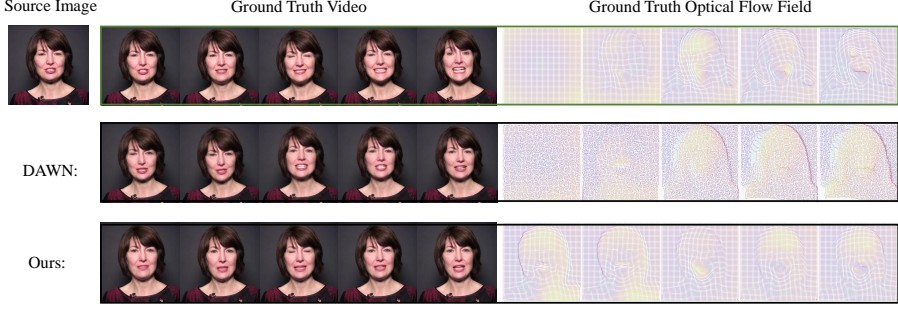

Figure 5: The visualization of the optical flow field of the Ground Truth video, DAWN, and our method. To enhance the visibility of jitters, pixels in the same row and column are connected with short lines.

Table 6: Quantitative study of user study. Participants evaluated videos on four aspects using a 1-5 scale: (1) L-Sync: lip-audio synchronization, (2) O-Nat: overall naturalness, (3) V-Qual: video quality, and (4) V-Stab: video stability. We calculated the mean score across all participants.

| Method | L-Sync | O-Nat | V-Qual | V-Stab |
|---|---|---|---|---|
| GT | 4.45 | 4.45 | 4.68 | 4.71 |
| Audio2Head | 2.53 | 2.39 | 2.57 | 3.08 |
| SadTalker | 2.48 | 1.75 | 1.93 | 2.61 |
| Hallo | 3.59 | 3.23 | 3.07 | 3.52 |
| EchoMimic | 3.60 | 3.33 | _3.79_ | 3.81 |
| AniTalker | 1.81 | 1.94 | 2.56 | 2.20 |
| DAWN | 3.21 | 2.92 | 3.61 | 2.49 |
| Hallo2 | **4.28** | _3.76_ | 3.62 | _4.04_ |
| Ours | _4.25_ | **3.88** | **4.16** | **4.34** |

### A.3 THE DEFINITION OF FLOW SMOOTHNESS

In this section, we present the definition of the Flow Smoothness (FS) metric. We first represent the optical flow field as a two-dimensional function with respect to coordinates $(x, y)$:

$$\boldsymbol{u}(x, y) = \big(u(x, y), v(x, y)\big), \tag{11}$$

where $u(x, y)$ and $v(x, y)$ denote the horizontal and vertical motion components at the two-dimensional coordinates $(x, y)$, respectively. Next, we define the gradient of the optical flow field as:

$$\nabla \boldsymbol{u} = \begin{pmatrix} \frac{\partial u}{\partial x} & \frac{\partial u}{\partial y} \\ \frac{\partial v}{\partial x} & \frac{\partial v}{\partial y} \end{pmatrix}. \tag{12}$$

Based on this, we further define the Flow Smoothness (FS) by Frobenius norm (denoted by the subscript "$F$") of the $\nabla \boldsymbol{u}$:

$$FS = \sqrt{\frac{1}{H \times W} \sum_{i=1}^{H \times W} \left( \left\| \nabla \boldsymbol{u}(x_i, y_i) \right\|_F^2 \right)}, \tag{13}$$

where $(x_i, y_i)$ is the coordinate of each sampled point.

### A.4 USER STUDY

To comprehensively evaluate our model's performance, we conducted a user study employing subjective metrics to compare our approach with previous methods. The study assessed generated videos across four dimensions: 1) L-Sync: Lip-audio synchronization; 2) O-Nat: Overall naturalness of generated results; 3) V-Qual: Overall video quality (e.g., presence of artifacts, abnormal color blocks, or color shifts; 4)V-Stab: Video stability (e.g., presence of flickering, jitters, or perturbations). We generated 10 test videos per method, with 20 participants scoring each on a scale of 1 to 5. The detailed scoring criteria for the user study are presented in Table 7. To establish a performance benchmark for the subjective metrics, we included ground truth videos among the test samples without informing the participants. According to the results in Table 6, our method outperforms all listed approaches across the four subjective metrics, ranking second only to ground truth videos in user evaluations. This demonstrates the exceptional quality of our generated results from the human user perspective.

### A.5 DISCUSSION

We analyze the visual quality improvements achieved by MDCF, which significantly reduces video jitters and enables more natural motions (e.g., blinking). The improvements stem from two key

Table 7: Scoring Criteria for Each Metric

| Metric | Scoring Criteria |
|---|---|
| **Lip Sync** | • 1: Completely inconsistent
• 2: Partially inconsistent (within 10 mismatch)
• 3: Generally consistent (within 5 mismatch)
• 4: Fairly consistent (within 3 mismatch)
• 5: Completely consistent |
| **Naturalness** | • 1: Very unnatural (obviously synthetic video)
• 2: Unnatural (sometimes appears significantly unnatural, with synthetic traces)
• 3: Generally natural (overall appears synthetic)
• 4: Fairly natural (generally consistent with real expressions, but occasionally shows synthetic traces)
• 5: Very natural (indistinguishable from real) |
| **Video Quality** | • 1: Very poor (many quality issues present)
• 2: Poor (many quality issues, greater than 10)
• 3: Average (some quality issues, greater than 5)
• 4: Good (few quality issues, less than 5)
• 5: Excellent (no quality issues) |
| **Frame Stability** | • 1: Very poor (many issues, severe shaking)
• 2: Poor (many issues present)
• 3: Average (some noticeable issues present)
• 4: Good (few issues present, minimal impact)
• 5: Excellent (no issues present) |

factors: (1) Compressed motion representation: MDCF transforms complex spatiotemporal distributions across numerous pixels into highly compressed features of a few tokens. For instance, blinking motions are abstracted into compact representations, substantially reducing learning complexity and improving motion quality. (2) Gaussian encoding robustness: The VAE's Gaussian encoding introduces beneficial noise that enhances decoder robustness and provides low-pass filtering of erroneous diffusion signals, effectively suppressing high-frequency interference for more stable video generation. These mechanisms enable MDCF to simultaneously improve generation efficiency, reduce computational overhead, and achieve superior motion modeling capabilities.

### A.6  LIMITATIONS AND FUTURE WORK

Some challenges still persist in our method. For example, the high compression ratio of MDCF leads to substantial downsampling of the generation space, complicating the construction of image conditioning for our diffusion model. In particular, the process of animating the speaker necessitates conditional features with robust segmentation capabilities, yet these capabilities are often constrained at low resolutions. This can cause the model to misidentify objects like headwear or hats as part of the background, consequently neglecting to generate motion for these areas. Consequently, a major future work direction is how to incorporate better segmentation capacity into our highly downsampled diffusion model.

A second limitation, which addresses the observation of smooth results that lack fine-grained local detail, likely stems from our reliance on MSE-based forward-KL losses (e.g., the diffusion loss

and the MDCF reconstruction losses). These losses are known to incentivize "mode averaging," which can inadvertently smooth out high-frequency signals and micro-expressions. In future work, we plan to explore methods to enhance these high-frequency motion signals without increasing the latent size. This includes potentially shifting to reverse-KL based optimization methods, such as incorporating reward models via reinforcement learning techniques (e.g., VideoDPO Liu et al. (2024a)). These approaches may allow the model to optimize directly for perceptual realism and detail, rather than pixel-wise averages.

## A.7 THE USE OF LARGE LANGUAGE MODELS (LLMS)

In this work, Large Language Models (LLMs) were used exclusively for language polishing. Specifically, their role was limited to grammar refinement, lexical optimization, and enhancing the fluency of expressions. LLMs did not participate in research ideation, core content development, or the writing of key sections. The authors bear full responsibility for all content of the paper, including its accuracy, originality, and compliance with academic ethics. LLMs are not eligible for authorship in this submission.

