# OpenReview forum: "Microscope: Efficient Diffusion with Two-Stage Dynamics Compression for High-Quality Talking Head Generation"
_ICLR.cc/2026/Conference — Submitted to ICLR 2026_

### Official Review · Reviewer_Vs1s · 2025-10-25

**Soundness:** 1
**Presentation:** 2
**Contribution:** 3
**Rating:** 4
**Confidence:** 3

**Summary:**

This paper introduces MICROSCOPE, a non-autoregressive diffusion framework for talking head generation, featuring a novel Microscope Dynamics Compression Framework (MDCF) and a Two-Phase Cascaded (TPC) training strategy. The proposed design achieves a 768× compression of pixel-level motion representations, enabling efficient long-term video generation (over 1600 frames at 512×512 resolution) using only 16GB of GPU memory. The approach seeks to address the limitations of autoregressive (AR) methods (error accumulation, slow inference) and prior non-autoregressive (NAR) models (excessive VRAM usage, flow jitter). Experimental results demonstrate superior FID/FVD and efficiency compared to DAWN, Hallo, and Audio2Head, with stable long-sequence generation and reduced motion artifacts.

**Strengths:**

- Impressive Computational Efficiency : Achieving over 1600-frame generation at high resolution on a single 16GB GPU is a notable technical feat. The proposed MDCF delivers strong compression (768×) without severe quality degradation, a clear step forward in resource-efficient diffusion models.
- Two-Phase Cascaded (TPC) Training Strategy : The separation of training between the Flow-aware Dynamics Extractor (FDE) and Latent Motion Auto-Encoder (LMAE) helps stabilize multi-stage compression and avoid gradient collapse. Ablations confirm substantial improvements in convergence and reconstruction quality when using TPC.
- Practical Long-Term Video Generation : Unlike most diffusion-based talking head models that are limited to short sequences (≤200 frames), MICROSCOPE demonstrates consistent temporal coherence across 1600+ frames. This establishes a solid baseline for scalable and efficient talking head synthesis.
- Comprehensive Evaluation : The paper presents quantitative, qualitative, and user studies, including a novel Flow Smoothness (FS) metric correlated with human judgment of motion stability. The combination of efficiency and quality benchmarks is thorough and convincing.
- Strong Engineering Contribution : The analogy to optical microscopes (multi-stage magnification) is well aligned with the multi-level compression principle. The proposed system is clearly articulated, reproducible, and accompanied by detailed experiments.

**Weaknesses:**

- Lack of Autoregressive (AR) Relevance : Although the paper positions itself against AR models, the presented architecture remains purely non-autoregressive. The claimed benefit for long-term temporal modeling is indirect—mainly due to compression and denoising—rather than actual sequence dependency learning.
- Limited Conceptual Novelty : The core contribution, MDCF, essentially extends hierarchical VAE-based compression (e.g., latent diffusion) to motion fields. While well-engineered, it feels incremental rather than conceptually groundbreaking.
- Overemphasis on Flow-Based Representation : The reliance on optical flow constrains expressiveness, leading to overly smooth or rigid facial motion. The model inherits typical flow-based limitations—difficulty capturing micro-expressions or out-of-plane movements.
- Unclear Core Message : The paper presents multiple intertwined technical elements—compression, cascaded training, flow filtering—but lacks a clear statement of which is the main contribution. The ablation studies confirm effectiveness but not the necessity or originality of each.
- Missed Connection to Prior Efficient Models : Given its emphasis on compression and inference speed, it would be logical to directly compare against efficiency-oriented baselines such as Audio2Head (GAN-based) or latent diffusion variants under equal conditions. The discussion of trade-offs between compression ratio and generation realism is also missing.

**Questions:**

- AR vs. NAR Trade-off : How does the proposed model ensure long-term temporal consistency without an autoregressive mechanism? Does compression alone suffice for maintaining coherence beyond 1600 frames?
- Main Takeaway : Among MDCF, TPC, and FS, which component represents the core novelty? How should future work position this framework—as a general latent diffusion scheme or a specialized talking head compressor?
- Motion Expressiveness : The results appear smooth but sometimes lack local detail. Is there a mechanism to enhance high-frequency motion signals without increasing the latent size?
- Baseline Relevance : Why not start from Audio2Head, which already offers a compact and efficient architecture for talking head generation? Would integrating MDCF into such a baseline further enhance its efficiency?
- Generalization and Compression Trade-off : The paper focuses on efficiency, but how well does MDCF generalize to unseen identities or different datasets (e.g., VoxCeleb2)? Does higher compression reduce generalization capacity?

**Details Of Ethics Concerns:**

The task involves generating human talking head videos, which inherently raises ethical considerations related to identity manipulation, consent, and potential misuse in deepfake applications. While the paper focuses on technical contributions, responsible research practices—such as data collection transparency, human subject consent, and safeguards against misuse—should be clearly discussed.

---

> ### Author Response · Authors · 2025-11-17
> **Author Response to Official Review by Reviewer Vs1s (1/4)**
>
> Thank you very much for your valuable and constructive comments on our paper. We are delighted that you recognize our efforts and contributions in terms of **computational efficiency** (generating over 1,600 frames on a 16GB GPU), the effectiveness of the **TPC training strategy**, **long video generation capability**, and **comprehensive evaluation**. Your critical feedback is extremely insightful, and we attach great importance to the rating of "Soundness: 1 (poor)". We believe this may stem from insufficient clarity in our elaboration on the **core novelty of MDCF** and **how it achieves long-term temporal consistency**. Your concerns (W1-W5) and questions (Q1-Q5) have pointed out clear directions for improvement. Below are detailed responses to your key concerns:
>
>
>
> ### **1. W1&Q1:**
>
> > Lack of Autoregressive (AR) Relevance; How ... ensure long-term temporal consistency without an autoregressive mechanism? Does compression alone suffice for maintaining coherence beyond 1600 frames?
>
> Thank you for your insightful comments. We acknowledge that the proposed method in this paper is a purely **non-autoregressive (NAR)** framework. Our comparison with autoregressive (AR) models aims to **address the inherent limitations of AR methods**: 1) **error accumulation** (particularly severe in long sequences) and 2) **slow inference speed**.
>
> * (a) This represents a critical misunderstanding: Our NAR framework performs "genuine sequence dependency learning."
>   It appears the reviewer may equate "sequence dependency learning" with "autoregressive (AR) learning." We argue that AR is merely one approach to sequence learning, whereas our NAR framework offers an alternative **global sequence learning paradigm** that is **more suitable for long videos**.
>
> * (b) How NAR enables global sequence dependency learning (in response to Q1):
>   Achieving non-autoregressive (NAR) coherence for long video sequences (3D) is an inherently difficult task. NAR generation for images (2D latnet space) is highly successful, in large part because the diffusion model operates on simpler 2D space. Similarly, one of the core contributions of our paper is to develope a compact and "easy-to-learn" latent space for long video dynamics via our MDCF framework. This extreme 768x compression is the **necessary prerequisite** that enables our diffusion model (DM) (equipped with a 3D-Unet backbone) to **process the entire set of dynamic latent variables** for the **full video sequence (1600+ frames) in parallel**. Furthermore, learning such a highly compressed representation makes the model's learning task significantly easier. The DM is thus tasked with learning a **global mapping** between the **complete** audio/pose sequences ($y_{1:N},\rho_{1:N}$) and the **complete** motion latent variable sequence ($\hat{z}_{1:N}$). At each denoising step, the model simultaneously observes and corrects **all** video frames. This parallel, global processing ensures coherence by mechanistically avoiding the error accumulation endemic to AR models.
>
>
> * (c) Compression is a necessary enabler of long sequence process.:
>   Compression itself does not guarantee coherence; it is a necessary prerequisite for handling "long-range sequence dependencies". The **core contribution** of our **768× compression** is that it enables processing the **entire** dynamic latent variables of 1600+ frames in VRAM for **parallel** denoising. Without MDCF, other NAR models would be forced to process short sequences (e.g., 200 frames) due to memory constraints, making them **fundamentally unable to handle** 1600-frame long-range dependencies. Thus, our advantages are **not** "indirectly attributed to compression" but **directly stem from** the "global sequence learning **enabled by** compression."
>
> * (d) **Why global NAR outperforms AR (coherence in 1600+ frames):**
>   Our global NAR method maintains consistency across 1600+ frames precisely because it **fundamentally avoids AR’s error accumulation**. For example, when predicting the 1600th frame, AR models rely on the (potentially erroneous) prediction of the 1599th frame. However, AR strategies typically use teacher-forcing during training, lacking error-correction capabilities. In contrast, our model generates the 1600th frame **in parallel** with all other frames, thereby mechanistically ensuring global temporal coherence and eliminating error accumulation.
>
> **Summary:** Our contribution is **not** "indirect." Instead, MDCF compression **enables** **genuine, global** NAR sequence dependency processing—a paradigm distinct from AR but better suited for long videos. By leveraging parallel processing and avoiding error accumulation, it ensures long-term consistency across 1600+ frames. We have clarified this logic more explicitly in the revised manuscript (line 46-49, 78).

---

> ### Author Response · Authors · 2025-11-17
> **Author Response to Official Review by Reviewer Vs1s (2/4)**
>
> ### **2. W2 & W4 & Q2**
>
> > MDCF, essentially extends hierarchical VAE-based compression (e.g., latent diffusion) to motion fields.
>
> We respectfully disagree with the view that MDCF is merely "incremental." We understand this confusion and will clarify the **fundamental differences** between MDCF and standard Latent Diffusion Models (LDMs):
>
> * (a) MDCF is not an LDM-VAE: Standard LDMs use a single-stage VAE to compress **pixel content**. In contrast, our MDCF is a carefully designed **two-stage cascaded framework** specifically for compressing **dynamics**.
>
> > W4 & Q2: MDCF, TPC, and FS, which  core novelty?
>
> * (b) MDCF Architecture (Core Novelty): The design of MDCF stems from our observation and analysis of the limitations of existing compression methods, rather than being a simple combination of them. To the best of our knowledge, a structure analogous to MDCF has not been studied in prior work.
>
>   * **First Stage (FDE - Objective Lens):** FDE is not a standard VAE. It is a flow-aware extractor (as shown in Section 3.1 and Fig. 1(a)) that captures **pixel-level** fine-grained motions (optical flow $f$ and mask $m$) with a **low compression ratio** (4×).
>
>   * **Second Stage (LMAE - Eyepiece Lens):** The LMAE (VAE) further compresses **the output of the first stage (i.e., dynamic representations $f, m$)**, rather than raw pixels.
>
> * (a) Necessity of TPC: As confirmed by our ablation study (Table 4, "w/o TPC"), this two-stage architecture cannot converge effectively through standard end-to-end training ("w/o TPC") (FID 12.75 vs. 7.84). Thus, the TPC training strategy represents a necessary methodological contribution that enables the MDCF architecture to function.
>
> * (b) Role of FS: FS is our proposed evaluation contribution, used to demonstrate that our method outperforms baselines in motion smoothness (as shown in Fig. 4b and Fig. 5)
>
> The core contribution is MDCF (a novel two-stage dynamic compression architecture), TPC is the necessary training strategy enabling its functionality, and FS is the metric for measuring the stability of its generated motion fields. We have strengthen this hierarchical relationship in the revised manuscript (line 24-27).
>
> > How should future work position this framework—as a general latent diffusion scheme or a specialized talking head compressor?
>
> Currently, this is a specialized talking head compressor. This is because talking head videos form a subspace on the manifold of video data distribution, featuring a simpler data distribution that enables relatively easy compression. We argue that MDCF theoretically possesses the ability to generalize to general video generation tasks, but further improvements and optimizations are required for its application in general videos. Therefore, we consider it appropriate to temporarily classify it as a talking head video compressor. In our future work, we also aim to explore its potential in general video generation models
>
>
>
>
> ### **3. W3：Overemphasis on Flow-Based Representation; Q3: Motion Expressiveness**
>
> > The reliance on optical flow constrains expressiveness, leading to overly smooth or rigid facial motion.
>
> We understand the reviewers' concerns regarding pure optical flow models; however, MDCF is not a "pure optical flow" model:
>
> - **FDE incorporates an Occlusion Map**: As described in Section 3.1 and Equation (1), the FDE not only extracts optical flow ( f ) but also an **occlusion map ( m )**. This map ( m ) is critical for handling **non-optical flow** motions, such as **newly exposed regions** (i.e., dis-occlusions) during actions like mouth opening or eye blinking. The image decoder (Equation 2) utilizes ( \otimes m ) to inpaint these regions.
>
> - **Role of LMAE(Q3)**: The VAE structure within LMAE (Equation 4) learns the **distribution** of motions through **Gaussian sampling**, rather than merely producing a deterministic optical flow field. Furthermore, its "low-pass filtering" property (detailed in Appendix A.5) is key to our high scores in the FS metric (Figure 4b) and user study (Table 5, V-Stab), effectively suppressing the jitter observed in previous work.
>
> **Conclusion**: Our model, through the combination of optical flow ( f ) + occlusion map ( m ) and VAE-based probabilistic modeling, exhibits stronger expressive power than pure optical flow methods.
>
>
> See the rebuttal for "Motion Expressiveness" in the next response.

---

> ### Author Response · Authors · 2025-11-17
> **Author Response to Official Review by Reviewer Vs1s (3/4)**
>
> We continue the rebuttal for "Motion Expressiveness":
>
> > The results appear smooth but sometimes lack local detail. Is there a mechanism to enhance high-frequency motion signals without increasing the latent size?
>
> We hypothesize that the loss of such high-frequency information may be related to forward-KL-based losses with MSE as the primary component (e.g., diffusion loss, the loss used to train MDCF). A key characteristic of these losses is their focus on learning the mean value (i.e., mode average), which can lead to the loss of high-frequency information. We believe that reinforcement learning methods based on reverse-KL (such as Video-DPO) or approaches incorporating reward models may yield better preservation of details. We plan to explore these methods in our future work.   We have incorporated the analysis of this weakness and the proposed future work into the "Limitations" and "Future Work" sections of the revised manuscript (line 863-870).
>
> ### **4.W5 & Q4 & Q5**
>
> > compare against efficiency-oriented baselines such as Audio2Head (GAN-based) or latent diffusion variants.  Why not start from Audio2Head.
>
> We appreciate the reviewer’s recommendation to compare with Audio2Head, and such a comparison has already been presented in our paper. Relevant comparisons are provided in **Table 1** and **Figure 4(a)** of the paper:
>
> * Table 1: Our method **significantly outperforms** Audio2Head across all **quality metrics** (e.g., FID: 11.22 vs 30.10; FVD-32: 60.28 vs 205.42). This demonstrates that while GAN-based methods are fast, they are far inferior to diffusion-based methods in terms of fidelity and stability.
>
> * Speed Comparison (Figure 4a): We acknowledge that Audio2Head (8.37s) is faster than our method (14.24s). However, this advantage stems from the inherent nature of GANs, which do not require iterative denoising—not from superior feature modeling. Our **contribution** lies in being the **faster high-quality diffusion model** (far outpacing other SOTA methods).
>
> * **Response to Q4 (Why not start from Audio2Head):**
>
>   1. We did not build upon Audio2Head because our goal is to address the **inefficiency** of **diffusion models**. Audio2Head is also a flow-based method and still suffers from the challenge of achieving high compression ratios. This issue is less prominent for GANs, but for diffusion models— which require iterative denoising—an uncompact latent space leads to more pronounced efficiency problems, which is precisely what we aim to solve. Additionally, low-compression representations relying solely on optical flow make training at high resolutions difficult, which is one of the reasons such methods (e.g., Audio2Head) are limited to 256×256 low-resolution generation. We have also compared our method with several LDM-based approaches, such as Hallo1, Hallo2 (recently added), and EchoMimic (Table 1, Figure 3). These experiments further validate the superiority of our method in both quality and efficiency.
>
>   2. Furthermore, GAN optimization is inherently challenging, unstable, and prone to mode collapse. Due to concerns about generation quality, we opted for diffusion as the generation algorithm.
>
> We have added explicit discussions in the revised manuscript to clarify the "speed-quality trade-off of GANs" and our positioning of "achieving the highest efficiency among diffusion-based SOTA methods."  (line 405-413)
>
> > W5 & Q5： Generalization and Compression Trade-off
>
> Thank you for your suggestion. We have further expanded **Table 2** by adding the compression performance of MDCF under a compression ratio of 192× (d/16, 16×16×3/4) to enhance the persuasiveness of the experiments.
>
> | Method      | FID↓  | FVD₁₆↓ | FVD₃₂↓ | LSE_C↑ | LSE_D↓ |
> | -| -| -| -| -| -|
> | GT| 0| 0| 0 | 8.30| 7.05|
> | FDE (d/4)   | 7.90| 21.18| 30.45| 7.92| 7.89|
> | FDE (d/8)   | 10.33| 35.07| 62.60| 6.89| 8.26|
> | VAE (d/32)  | 24.26 | 62.84| 100.61| 7.64| 7.52|
> | MDCF (d/16) | 7.85| 21.31| 30.40| 7.95| 7.38|
> | MDCF (d/32) | 7.84| 20.84| 29.46| 7.92| 7.36|
>
> Table 2 shows that under the same 768× compression ratio (i.e., $d/32$), the performance of the **single-stage VAE ("VAE (d/32)")** ** degrade severely** (FID 24.26, FVD 100.61). In contrast, our **two-stage MDCF ("MDCF (d/32)")** maintains extremely high quality at the **same compression ratio** (FID 7.84, FVD 29.46). This strongly demonstrates that our MDCF architecture is **not "incremental"** but a **necessary** design to achieve 768× compression.  Additionally, we find that the performance of MDCF with a 192× compression ratio is close to that with 768×, indicating that our method maintains good performance within a reasonable range of parameters, and there is not necessarily a strict trade-off between compression ratio and reconstruction quality. However, we acknowledge that further increasing the compression ratio may require additional model structure designs or training optimizations to achieve desirable performance, which we plan to explore in future work.

---

> ### Author Response · Authors · 2025-11-17
> **Author Response to Official Review by Reviewer Vs1s (4/4)**
>
> > Q5: how well does MDCF generalize to unseen identities or different datasets (e.g., VoxCeleb2)?
>
> We selected HDTF as the primary dataset because one of our core objectives is to address the challenges of **high-resolution (512x512)** and **long-video (1600+ frames)** generation. HDTF serves as an ideal platform to validate this advantage, as many SOTA methods fail under this setting due to VRAM constraints or error accumulation (as shown in Table 3 of the paper). We fully agree that testing on additional datasets enhances the persuasiveness of our experiments. Therefore, we have supplemented experiments on the VoxCeleb2 dataset and use sota methods hallo and hallo2 for comparison.
>
> |        | fid   | Fvd 16 | Fvd 32 | sync-C | sync-D | CSIM  |
> | ------ | ----- | ------ | ------ | ------ | ------ | ----- |
> | hallo  | 34.59 | 323.65 | 545.94 | 4.72   | 9.23   | 0.610 |
> | hallo2 | 24.62 | 233.45 | 389.77 | 4.93   | 9.50   | 0.643 |
> | ours   | 18.54 | 207.83 | 288.04 | 5.02   | 9.39   | 0.662 |
>
> We deployed the MDCF model trained exclusively on HDTF to drive test videos from VoxCeleb2. Notably, even without any exposure to VoxCeleb2 data during training, our method still achieves competitive performance compared to previous works. This demonstrates that our approach—especially the FDE stage—learns universal, pixel-based motion patterns rather than overfitting to the data distribution of HDTF.  We have add the extra experiment and analysis in line 412-425. We hope this addresses your concerns and further strengthens our paper.
>
> ### **5. Ethics Concerns**
>
> We fully agree with the reviewers' comments. This is both technically necessary and ethically responsible.
>
> We have added a dedicated section entitled "Ethics Statement" in the revised manuscript (line 540-556). Within this section, we have addressed the following points:
>
> 1. The datasets we used (HDTF) and the additionally incorporated VoxCeleb2 are publicly available research datasets.
>
> 2. Acknowledgment of the potential for this technology—like all talking head technologies—to be misused for "deepfakes."
>
> 3. Emphasis on its positive applications, such as digital humans, virtual assistants, education, and assistive functionalities.
>
> 4. A statement of our commitment to responsible research, along with a note that future work may explore watermarking or detection mechanisms.
>
> ----
> We would like to once again express our gratitude to the reviewers for their valuable time. We believe that through the above clarifications—particularly the elaboration on the novelty of MDCF's core architecture—as well as the supplementary **generalization experiments (on VoxCeleb2)** and more detailed analyses, the "soundness" of this paper will be strongly substantiated. We hope that the revised manuscript will meet the acceptance criteria of ICLR.

---

> ### Author Response · Authors · 2025-11-24
> **We hope that our response addresses your concern**
>
> Dear reviewer Vs1s,
>
> We greatly appreciate the time you've invested in reviewing our response. Having submitted our rebuttal, we are eager to know if our response has addressed your concern. As the end of the rebuttal phase is approaching, we look forward to hearing from you for any further clarification that you might require.
>
> Best, Submission 9326 authors

---

### Official Review · Reviewer_v3Mf · 2025-10-30

**Soundness:** 3
**Presentation:** 2
**Contribution:** 2
**Rating:** 4
**Confidence:** 4

**Summary:**

The paper proposes a novel non-autoregressive talking head generation framework that compresses motion dynamics, significantly reducing computational resource. This paper introduces two-stage pipeline, Microscope Dynamics Compression Framework (MDCF), which first trains the Flow-aware Dynamics Extractor (FDE) to capture motions and then optimizes the Latent Motion AutoEncoder (LMAE). A Two-Phase Cascaded (TPC) training scheme and an image-guided consistency (IGC) loss stabilize training. Advantages including 768x compression and long videos (more than 1600 frames) at 512x512 on a single 16GB GPU are practical and interesting.

**Strengths:**

1. The paper is well written and clear. It is also technically sound and grounded.
2. Two-stage dynamics compression makes sense. The paper also validated the effectiveness of its architectural choices by extensive ablations.
3. The method is effective and resource-friendly.

**Weaknesses:**

1. Narrow evaluation dataset
Train and eval are confined to HDTF. Cross-dataset tests (e.g., LRS3, VoxCeleb2, CelebV..) are needed to support generalization.

2. Small / under-reported user study
Only 10 participants and 6 test videos per method  seems not that reliable. Extensive user study is needed. (More participants and more test cases needed)

3. FS metric concerns/fairness
FS is defined on optical flow gradients. Since your method learns flow and reconstructs,  FS may favor your method. Moreover, isn't there any extreme case where an overly smooth (even wrong) flow achieves a low FS, since the metric measures smoothness rather than motion correctness?

4. Lip sync accuracy comparisons
In the Table1 (Main quantitative comparison on HDTF), the performance on lip sync accuracy is not optimal relative to other SOTA models (also, several baselines appear out-of-date. I think more recent and powerful models should be compared. e.g., Hallo2, Hallo3, OmniSync, StableAvatar.. ). More importantly, could you clarify which factor limits lip-sync: (i) does a flow-only motion representation fail to capture local mouth-region? or (ii) is audio conditioning or alignment limited? Relatedly, with flow prediction, do you observe mouth-area artifacts (e.g., teeth not rendered correctly)?

5. Missing identity-related metric
The paper reports FID, FVD and lip-sync metrics (LSE-D, LSE-C), but no identity-similarity metrics provided.

**Questions:**

I would appreciate responses to the questions in the Weaknesses section

---

> ### Author Response · Authors · 2025-11-17
> **Author Response to Official Review by Reviewer v3Mf (1/3)**
>
> Thank you very much for your meticulous review of our paper. We are delighted that you recognize the **clarity**, **technical soundness**, **effectiveness of the two-stage compression design**, and **comprehensive ablation studies** of our work. Your comment that our method is "efficient and resource-friendly" aligns perfectly with the core goal of this research. The weaknesses you identified regarding **Evaluation** (W1-W5) are extremely insightful and constructive. We agree that the current version has room for improvement in terms of generalization, user studies, metric fairness, and baseline comparisons. We greatly appreciate you pointing out these issues and supplement all missing experiments.
>
> Below are detailed responses to your key concerns:
>
> ###  **1, W1:**
> > Narrow evaluation dataset
>
> We selected HDTF as the primary dataset because one of our core objectives is to address the challenges of **high-resolution (512x512)** and **long-video (1600+ frames)** generation. HDTF serves as an ideal platform to validate this advantage, as many SOTA methods fail under this setting due to VRAM constraints or error accumulation (as shown in Table 3 of the paper). We fully agree that testing on additional datasets enhances the persuasiveness of our experiments. Therefore, we have supplemented experiments on the VoxCeleb2 dataset and use sota methods hallo and hallo2 for comparison.
>
> |       | fid   | Fvd 16 | Fvd 32 | sync-C | sync-D | CSIM   |
> |-------|-------|--------|--------|--------|--------|--------|
> | hallo| 34.59 | 323.65 | 545.94 | 4.72   | 9.23   | 0.610 |
> | hallo2| 24.62 | 233.45 | 389.77 | 4.93   | 9.50   | 0.643 |
> | ours  | 18.54 | 207.83 | 288.04 | 5.02   | 9.39   | 0.662 |
>
>
> We deployed the MDCF model trained exclusively on HDTF to drive test videos from VoxCeleb2. Notably, even without any exposure to VoxCeleb2 data during training, our method still achieves competitive performance compared to previous works. This demonstrates that our approach—especially the FDE stage—learns universal, pixel-based motion patterns rather than overfitting to the data distribution of HDTF.  We have add the extra experiment and analysis in line 412-425. We hope this addresses your concerns and further strengthens our paper.
>
>
> ###  **2. W2:**
> > Extensive user study is needed
>
> Thank you for your correction. The scale of our user study was primarily referenced from previous work [1]. However, we acknowledge that the current user study is limited in scale, which may prevent us from drawing statistically robust conclusions. Due to resource constraints, we are unable to conduct large-scale user studies involving hundreds or thousands of participants. Therefore, within our capacity, we have expanded the test scope as much as possible: we additionally invited 10 users (totaling 20 participants) and add 4 more test videos (resulting in 10 test videos in total). The results are as follows (Table 6):
>
> | Method      | L-Sync | O-Nat | V-Qual | V-Stab |
> |-------------|--------|-------|--------|--------|
> | GT          | 4.45   | 4.45  | 4.68   | 4.71   |
> | Audio2Head  | 2.53   | 2.39  | 2.57   | 3.08   |
> | SadTalker   | 2.48   | 1.75  | 1.93   | 2.61   |
> | Hallo       | 3.59   | 3.23  | 3.07   | 3.52   |
> | EchoMimic   | 3.60   | 3.33  | 3.79   | 3.81   |
> | AniTalker   | 1.81   | 1.94  | 2.56   | 2.20   |
> | DAWN        | 3.21   | 2.92  | 3.61   | 2.49   |
> | Hallo2      | 4.28   | 3.76  | 3.62   | 4.04   |
> | Ours        | 4.25   | 3.88  | 4.16   | 4.34   |
>
> We observed that despite some fluctuations in the results, our method still acchieve comparable or better performance to other approaches in the user study (including hallo2), achieving visual perception closer to the GT videos. In future work, we aim to further develop an MLLM-based automatic scoring system. Leveraging the characteristic that these models are highly aligned with human preferences, we hope to achieve accuracy comparable to hundred/thousand-scale user studies at a lower cost.
>
>
> [1] Liu, Tao, et al. "Anitalker: animate vivid and diverse talking faces through identity-decoupled facial motion encoding." Proceedings of the 32nd ACM International Conference on Multimedia. 2024.

---

> > ### Author Response · Authors · 2025-11-17
> > **Author Response to Official Review by Reviewer v3Mf (2/3)**
> >
> > ### **3. W3:**
> > > FS metric concerns/fairness
> >
> > This is an astute observation. We would like to clarify the fairness and validity of the FS metric from two perspectives:
> >
> > - **Regarding Fairness**: First, the FS metric (defined in Appendix A.3) extracts optical flow from the **final generated video frames** and calculates gradients, rather than directly evaluating the latent variable $f$ within our model. Second, our primary baseline for comparison, DAWN, is also a NAR method based on optical flow representation. Thus, this constitutes a *comparison within the same category*, evaluating which of the two optical flow-driven models produces *less jitter in the final video*, with no bias involved.
> >
> > - **Regarding Over-smoothing/Errors**: This is a reasonable theoretical concern. However, our experiments (Fig. 4b) demonstrate that we have avoided such issues. Our goal is **not** to achieve the lowest FS value, but rather the FS value **closest to that of the Ground Truth (GT)** (see lines [insert line numbers]).
> >
> > As shown in the data from Fig. 4(b): The FS value of GT (real video) is **0.0703**; DAWN’s FS value is **0.1211** (a higher value indicates abundant abnormal gradients, i.e., jitter); and our FS value is **0.0803**. This indicates that our method has only **effectively eliminated the abnormal high-frequency jitter in DAWN**, restoring motion smoothness to a natural level **close to GT**, rather than being "over-smoothed" to an unrealistic low value. Meanwhile, our other metrics (e.g., FID/FVD in Table 1) also validate the correctness of the motion.
> >
> > ###  **4. W4:**
> > > Lip sync accuracy, baseline concers,  clarify which factor limits lip-sync.
> >
> > - (a) We acknowledge that our LSE scores (Table 1) are not the *optimal* among all SOTA methods. However, our scores (e.g., $LSE_C$ 7.33, $LSE_D$ 7.85) are **highly competitive** and nearly on par with other SOTA methods. Our core contribution lies in significantly improving efficiency while maintaining SOTA-level quality (our generation speed is over 40 times that of Hallo). Additionally, it should be clarified that **both Hallo and the supplementary Hallo2 incorporate HDTF into their training sets [1]**. Since they have not reported the division of HDTF into training and test sets, we were unable to adopt the same partitioning. Consequently, from a probabilistic perspective, our test set very likely includes data from the training sets of these two methods, which may introduce a certain degree of bias in their favor.
> >
> > - (b) Outdated baselines: Thank you for this reminder. Such methods typically utilize pre-trained  video generation models (e.g., AnimateDiff ) and finetuned large-scale data (**over 100+ hours video**), whereas our approach is trained from scratch (**with only 10-20 hours video**). Strictly speaking, such comparisons are not entirely fair, which is why we initially only included EchoMimic and Hallo as representative examples of this category. However, we agree that adding more baselines enhances the persuasiveness of our experiments. We have supplemented comparisons with Hallo2 to provide a more comprehensive evaluation in revised paper, Table 1. As shown in the result, our method also acchieve comparable or better results to the hallo2.
> >
> > |  | fid    | Fvd 16 | Fvd 32 | sync-C | sync-D | CSIM |
> > |-|-|-|-|-|-|-|
> > | hallo2 | 19.10| 113.30 | 164.58 | 7.379  | 7.697  | 0.789  | 0.789  |
> > | ours-256 | 11.22| 44.6 |60.28  | 7.33   | 7.85   | 0.791  | 0.789  |
> > | ours-512 | 12.57| 51.02| 65.4   | 7.14   | 7.95   | 0.806  | 0.789  |
> >
> > - (c) Limiting factors: We do not consider the current FDE as the most significant bottleneck for performance. As shown in Table 2, when testing the FDE solely, the reconstruction loss of FDE is relatively minimal; the primary bottleneck lies in the generative capability of the diffusion backbone. We believe that model scale and training data are critical limiting factors. Due to resource constraints, our model has 200-300M parameters, whereas methods with superior lip-sync performance typically have over 700M parameters (e.g., Hallo, Hallo2) and utilize significantly more training data and computational resources. In future work, we aim to further scale up our method (both model and data), especially the diffusion backbone, to explore the performance upper bound of the framework.
> >
> > - (d) Mouth artifacts: Our FDE does not rely solely on optical flow. As described in Section 3.1, it simultaneously predicts optical flow $f$ and mask $m$. The mask $m$ is specifically designed to **repair occluded or newly exposed regions** (e.g., teeth visible when the mouth opens) in the image decoder. While it may not be perfect in extreme scenarios, this design explicitly addresses and mitigates artifact issues inherent to pure flow-based models. We aim to address this comprehensively in future work.
> >
> > [1] Cui, Jiahao, et al. "Hallo2: Long-duration and high-resolution audio-driven portrait image animation." arXiv preprint arXiv:2410.07718.

---

> ### Author Response · Authors · 2025-11-17
> **Author Response to Official Review by Reviewer v3Mf (3/3)**
>
> ###  **5. W5:**
> > no identity-similarity metrics provided.
>
> We agree that preserving the identity of the source image is crucial when evaluating the generation quality of Talking Heads. Therefore, we have supplemented the CSIM metric. The compeleted results are add to the revised paper, Table 1. It can be observed that our method has also achieved performance comparable to SOTA methods, including Hallo2, in identity preservation. We hope this addresses your concern and makes the experiments more persuasive.
>
>
> | Method      | CSIM ↑ |
> |-------------|--------|
> | Audio2Head  | 0.705  |
> | SadTalker   | 0.767  |
> | Hallo       | 0.709  |
> | EchoMimic   | 0.731  |
> | AniTalker   | 0.725  |
> | DAWN      | 0.790  |
> | Hallo2    | 0.789  |
> | Ours-256    | 0.791  |
> | Ours-512    | 0.806  |

---

> ### Author Response · Authors · 2025-11-24
> **We hope that our response addresses your concern**
>
> Dear reviewer v3Mf,
>
> We greatly appreciate the time you've invested in reviewing our response. Having submitted our rebuttal, we are eager to know if our response has addressed your concern. As the end of the rebuttal phase is approaching, we look forward to hearing from you for any further clarification that you might require.
>
> Best, Submission 9326 authors

---

### Official Review · Reviewer_PMYf · 2025-11-01

**Soundness:** 2
**Presentation:** 2
**Contribution:** 3
**Rating:** 4
**Confidence:** 4

**Summary:**

This paper proposed a method to generation talking head videos based on diffusion video generation model. The core idea is to use a two-stage cascaded pipeline to disentangle the facial latent motion prediction and face reconstruction based on motion. The overall design achieves significant runtime speed up and VRAM consumption reduction.

**Strengths:**

- The proposed MDCF framework divides the talking face video generation into two cascaded stages. The first flow-aware dynamic extractor disentangle the face dynamics into facial identity and flow motion. The second latent motion AE further compress the flow motion dynamics into low-dim latent for later faster computation.
- The proposed image guided consistency loss and KL regularization helps the training in mitigating the error accumulation for multi-stage design.

**Weaknesses:**

- Though the high level idea of FDE stage is clear, it is not clear what is the details of it. Specially, how it “aggregates nearby pixels into patches”? What is “a slight compression of the dynamics representation”?
- How is the model performed when the facial dynamics are not just warping? In the supplementary video, the overall lip/jaw dynamics looks good for warping motions but fails dramatically on lip shape change motions, like /o/, /th/, /b/p/m/ sounds. The issue might come from the FDE stage which is based on flow-based warping.

**Questions:**

- In Section 4.3, it is claimed that the proposed method, especially FDE stage, out performs LIA. It is not clear to me why LIA achieves 4x higher FID/FVD scores in Table 4. It would be helpful to provide comparison for their failure cases.
- How is the eye blinking animation generated in the out of distribution test video, while no eye blinking animation in comparison video?

---

> ### Author Response · Authors · 2025-11-17
> **Author Response to Official Review by Reviewer v3Mf (1/2)**
>
> We sincerely appreciate your meticulous review and valuable feedback on our paper. We are pleased that you recognize the contributions of our proposed MDCF framework in phased compression, reducing VRAM consumption, and improving inference speed, as well as the effectiveness of IGC and KL losses in mitigating error accumulation during multi-stage training. In response to the concerns and questions you raised, we provide detailed explanations and rebuttal. We have supplemented all necessary experiments and descriptions in the revised version to fully address your considerations.
>
>
> ###  **1. W1:**
> > how it “aggregates nearby pixels into patches”? What is “a slight compression of the dynamics representation”?
>
>
> We apologize for the confusion caused by our description. This is indeed an aspect of the paper that can be improved, and we provide a clearer explanation here:
>
> - "aggregates nearby pixels into patches"
>   What we intended to convey is the role of the **image encoder (Img Encoder)** within FDE. In our framework (as shown in Figure 1(a)), the warping operation is not performed in the original pixel space but in a **downsampled latent space**. The image encoder $\mathcal{E}(x_{src})$ maps the high-resolution input image $x_{src}$ to a downsampled latent feature space. This encoding process (typically via convolutional or downsampling layers) inherently aggregates spatially adjacent pixel information into latent feature vectors. Thus, "aggregation" refers to the **intrinsic property** of this encoder’s downsampling, rather than an independent patching step.
>
> - "A slight compression of the dynamics representation"
>   Here, "compression" refers to the **spatial downsampling rate** introduced by the image encoder in the FDE stage. As you are aware, the core of FDE lies in flow-based warping. If the downsampling rate at this stage is set **too high** (i.e., "heavy compression"), the resolution of the latent space becomes excessively low, leading to a **severe degradation** in warping accuracy and loss of critical local motion details (e.g., subtle lip movements). Therefore, the FDE stage **must** use a small downsampling rate (i.e., "slight compression," e.g., 4x) to ensure the precision of motion modeling. This is precisely the original motivation behind MDCF’s design: FDE (the first-stage objective lens) handles "slight compression" to preserve motion accuracy, while LMAE (the second-stage eyepiece) further compresses spatial redundancy on this basis. Their cascading enables efficient compression at an overall ratio of 768x without sacrificing quality.
>
> We have revise these descriptions in Section 3.1 (line line 182-189) of the revised paper for better clarity.
>
>
> ### **2. W2:**
> > facial dynamics are not just warping; fails on lip shape change motions, like /o/, /th/, /b/p/m/ sounds
>
>
> Thank you for this astute and critical observation.
>
> We acknowledge that this is indeed a **shared limitation across all flow-based modeling methods**. Flow fields inherently excel at simulating pixel "displacements" and "distortions" (warping), but they cannot perfectly model regions with topological changes—such as a closed lip transforming into an "opening" (/o/) or the exposure/occlusion of teeth/tongues.
>
> **Our solution (Occlusion Map):** Anticipating this issue, our FDE framework **does not solely** rely on the flow field $f$. As illustrated in Figure 1(a), FDE simultaneously predicts both the flow field $f$ and an occlusion map $m$. The flow field $f$ handles motions amenable to warping (e.g., rotations of the cheek or jaw). The occlusion map $m$, by contrast, identifies regions where warping fails (i.e., "occluded areas" or "newly exposed regions," such as the interior of an /o/-shaped mouth). Finally, the image decoder $\mathcal{D}$ leverages the occlusion map $m$ to "in-paint" or "redraw" these regions.
>
> The design of the occlusion map $m$ has significantly mitigated the limitations of pure flow-based methods. We acknowledge that the model may not be perfect for certain extreme articulations—a universal challenge in current flow-based approaches. We aim to address this comprehensively in future work.

---

> > ### Author Response · Authors · 2025-11-17
> > **Author Response to Official Review by Reviewer v3Mf (2/2)**
> >
> > ### **3. Q1**
> > > LIA failure cases.
> >
> > LIA adopts a **disentanglement strategy** between motion and identity. While theoretically elegant, this disentanglement is **extremely challenging** in practice and prone to causing "performance bottlenecks." The failure cases of LIA are mainly manifested as: visually perceptible overall blurriness (loss of high-frequency features), incorrect driving, and loss of fine lip movements. These issues directly lead to the absence or distortion of image/video features, resulting in inferior FID/FVD and lip-sync metrics.
> >
> > Due to concerns about such disentanglement difficulties and degraded image quality, our MDCF **intentionally avoids** this high-risk disentanglement approach. Instead, it employs a more robust and direct **pixel-level motion modeling** (i.e., flow fields + masks of FDE), prioritizing the **high fidelity** and **robustness** of video reconstruction.
> >
> > The following link ([MDCF_cmp_LIA](https://anonymous.4open.science/r/MDCF-0D83/MDCF_cmp_LIA.mp4)) provides further demonstrations of LIA’s failure cases. We hope these materials will resolve any confusion you may have regarding the experimental results.
> >
> >
> >
> >
> >
> >
> > ### **4. Q2:**
> > > Regarding Eye Blinking Animation
> >
> > **Eye blinking is a conditional input to the model**: Our model **can indeed** generate natural eye blinking animations. Blink signals, alongside head poses, are fed as conditions into the core diffusion model (DM).
> >
> >   - **During training**: We extract **realistic** head poses and blink signals from authentic video clips to supervise the diffusion model’s generation process.
> >   - **During inference**: An additionally trained **PBNet (Pose & Blink Network)** describe in line 327. This network takes **audio as input** and outputs **synthesized** head poses and **blink sequences**. Built on a VAE architecture, it introduces controlled stochasticity in generation.
> >
> > Regarding the absence of blinks in the "comparison video" you noted: For the first example, no blinks were generated within the 8-second window due to the stochasticity of PBNet, which aligns with the sparse nature of real blinking. We have re-inferred the first example with a different random seed, extended the video length, retained the other two examples unchanged, and annotated the timestamps of blinks. Please further refer to the link [our_method_mark_blink](https://anonymous.4open.science/r/MDCF-0D83/our_method_mark_blink.mp4).

---

> ### Author Response · Authors · 2025-11-24
> **We hope that our response addresses your concern**
>
> Dear reviewer PMYf,
>
> We greatly appreciate the time you've invested in reviewing our response. Having submitted our rebuttal, we are eager to know if our response has addressed your concern. As the end of the rebuttal phase is approaching, we look forward to hearing from you for any further clarification that you might require.
>
> Best,
> Submission 9326 authors

---

### Official Review · Reviewer_QWbo · 2025-11-01

**Soundness:** 3
**Presentation:** 4
**Contribution:** 3
**Rating:** 4
**Confidence:** 2

**Summary:**

This paper proposes MICROSCOPE, an efficient diffusion-based framework for audio-driven talking head generation. The key contribution is the Microscope Dynamics Compression Framework (MDCF), which combines a Flow-aware Dynamics Extractor (FDE) and a Latent Motion Auto-Encoder (LMAE) to achieve a 768× compression ratio of motion dynamics while maintaining fidelity. A Two-Phase Cascaded (TPC) training strategy and Image-Guided Consistency (IGC) loss stabilize multi-stage optimization. MICROSCOPE enables long, high-resolution video synthesis 1,and shows strong results on HDTF, outperforming previous non-autoregressive models in both quality and efficiency.

**Strengths:**

1. The two-stage compression framework (FDE–LMAE) is coherent, empirically supported, and yields clear memory and latency benefits.
2. Clearly motivated by efficiency and scalability issues in diffusion-based talking head generation.

**Weaknesses:**

1. The overall novelty is moderate, as the framework mainly combines existing latent compression and staged training techniques.
2. Evaluation is limited to HDTF. Generalization to other datasets (e.g., VoxCeleb2 or in-the-wild) remains uncertain.
3. The approach appears to rely on carefully tuned downsampling factors and loss weights, but the rationale behind these design choices is not clearly explained. Providing more insight or adaptivity in how these parameters are selected would make the framework more convincing.

**Questions:**

1. Could the authors comment on how well the model generalizes to more diverse or in-the-wild datasets beyond HDTF?
2. How were the downsampling factors and loss weights chosen in practice? Have the authors observed any trade-offs between the high compression ratio (e.g., 768×) and fine-grained motion or expression fidelity?

---

> ### Author Response · Authors · 2025-11-17
> **Author Response to Official Review by Reviewer QWbo (1/2)**
>
> We sincerely appreciate your meticulous review and insightful feedback on our work. You have accurately summarized the core contributions of our MDCF framework and recognized our efforts in addressing efficiency and scalability challenges. In response to the valuable questions raised in your "Weaknesses" and "Questions" sections, we have prepared comprehensive supplements and clarifications:
>
> ### **1. w1: regarding the novelty of framework**
>
> Thank you for raising this point. We acknowledge that FDE and LMAE are established technical concepts in their respective domains.  However, our core novelty does not lie in simply "combining" these two techniques, but rather in our profound insight into the limitations of each method when applied in isolation, and the subsequent proposal of the cascaded complementary framework (MDCF) based on this understanding.
>
> Our observations, as illustrated in Table 2 of the paper, reveal two critical bottlenecks:
>
> - Bottleneck of flow-based methods (FDE): FDE relies on the assumption of local motion consistency. At low compression rates (e.g., 4x), it captures motion with high fidelity (as shown in the "FDE (d/4)" row). However, when used alone for heavier compression (e.g., "FDE (d/8)"), its core assumption is violated, leading to a drastic drop in fidelity (FVD surges from 30.45 to 62.60).
>
> - Bottleneck of standard VAE (LMAE): Using a standalone VAE to achieve high compression in a single step (e.g., 768x, see "VAE (d/32)" row) results in catastrophic information loss, with reconstruction quality (FVD 100.61) rendered entirely unusable for subsequent generative tasks.
>
> Our MDCF framework is specifically designed to circumvent both bottlenecks simultaneously. It is not a mere "addition" but an elegant complementary collaboration analogous to a "microscope":
> - FDE (objective lens): Handles the first stage of "high-fidelity, low-compression" (4x) to ensure the accuracy of warping transformations and preserve pixel-level fine-grained motion.
> - LMAE (eyepiece): Manages the second stage of "high-magnitude, high-efficiency" compression, operating not on raw pixels but on the preliminarily condensed motion information output by FDE.
>
> This two-stage "fidelity-first, then compression" strategy represents a non-trivial design, enabling us to achieve an overall compression ratio of 768x which cannot acchieved by both individual methods.
>
>
> ### **2. W2 and Q1: regarding the limitation of dataset, Generalization ability**
>
> We selected HDTF as the primary dataset because one of our core objectives is to address the challenges of **high-resolution (512x512)** and **long-video (1600+ frames)** generation. HDTF serves as an ideal platform to validate this advantage, as many SOTA methods fail under this setting due to VRAM constraints or error accumulation (as shown in Table 3 of the paper). We fully agree that testing on additional datasets enhances the persuasiveness of our experiments. Therefore, we have supplemented experiments on the **VoxCeleb2** dataset and use sota methods hallo and hallo2 for comparison.
>
> |       | fid   | Fvd 16 | Fvd 32 | sync-C | sync-D | CSIM   |
> |-------|-------|--------|--------|--------|--------|--------|
> | hallo| 34.59 | 323.65 | 545.94 | 4.72   | 9.23   | 0.610 |
> | hallo2| 24.62 | 233.45 | 389.77 | 4.93   | 9.50   | 0.643 |
> | ours  | 18.54 | 207.83 | 288.04 | 5.02   | 9.39   | 0.662 |
>
>
> We deployed the MDCF model trained exclusively on HDTF to drive test videos from VoxCeleb2. Notably, even without any exposure to VoxCeleb2 data during training, our method still achieves competitive performance compared to previous works. This demonstrates that our approach—especially the FDE stage—learns universal, pixel-based motion patterns rather than overfitting to the data distribution of HDTF.  We have add the extra experiment and analysis in line 412-425. We hope this could address your concerns and further strengthen our paper.

---

> > ### Author Response · Authors · 2025-11-17
> > **Author Response to Official Review by Reviewer QWbo (2/2)**
> >
> > ### **3. W3 & Q2: regarding the selection of hyperparameters, trade-offs between compression ratio and fidelity**
> >
> > > rely on carefully tuned downsampling factors
> >
> > This is a critical question that touches upon the core of our design. The selection of these parameters is not arbitrary "tuning" but is based on our in-depth understanding of the framework and empirical evidence.
> >
> > - Downsampling rates (FDE: 4x, LMAE: 8x):
> >   - FDE (4x): The downsampling rate at this stage must be small. As highlighted in our response to W1, once FDE’s compression ratio increases, motion fidelity is severely compromised. Thus, the choice of 4x ($\frac{H}{4} \times \frac{W}{4}$) is an empirically validated, "safe and necessary" compression that balances feature extraction and motion accuracy.
> >   - LMAE (8x): During initial experiments, we observed no significant downsampling rate-quality tradeoff when LMAE’s downsampling rate ranged from 4x to 8x as we shown in the table below. However, practical tests revealed quality degradation when the resolution exceeded 16x. Therefore, we ultimately selected 8x downsampling for LMAE to balance quality and efficiency. In summary, 4x (FDE) prioritizes "fidelity," while 8x (LMAE) prioritizes "efficiency."
> >
> > | Method        | conf         | dist  | fid   | fvd16  | fvd32  |
> > |--------------|-------|-------|--------|--------|--------|
> > | MDCF(LMAE 8x)   | 7.92  | 7.36  | 7.84   | 20.84  | 29.46  |
> > | MDCF(LMAE 4x)   | 7.95  | 7.38  | 7.85   | 21.31  | 30.40   |
> >
> >
> > we have incoporate the extra experiment in Table 2, and discussion in line 511-514.
> >
> > > rely on carefully tuned loss weigths
> >
> > Loss weights (Sec 4.1: $\lambda_{KL}=10^{-4}, \lambda_{MSE}=1, \lambda_{IGC}=1$):
> >   - $\lambda_{KL}=10^{-4}$: This is a **standard** weight in VAE training, serving to lightly regularize the model to prevent posterior collapse without compromising reconstruction quality.
> >   - $\lambda_{MSE}=1, \lambda_{IGC}=1$: We set equal weights (1.0) for MSE (feature-level reconstruction) and IGC (image-level reconstruction) because both are equally critical:
> >
> > These weights represent a straightforward choice to maintain balance within a reasonable range (e.g., KL weight being much smaller than reconstruction terms), rather than the result of extreme "fine-tuning." We have incoporate these analysis in line 343-347, section 4.1.
> >
> >
> >
> > > trade-offs between the high compression ratio (e.g., 768×) and fine-grained motion or expression fidelity
> >
> > Theoretically, there is indeed a trade-off, as fewer tokens significantly increase the difficulty of reconstructing fine motion features. However, our method is specifically designed to mitigate this issue. As discussed above, we have analyzed the impact of hyperparameters and selected values that balance efficiency and reconstruction quality. Overall, within a reasonable parameter range (e.g., compression ratios between 196x and 768x), there is no noticeable trade-off between compression ratio and reconstruction accuracy for the MDCF framework. Nevertheless, compression ratios higher than 768x do lead to a degradation in reconstruction precision.

---

> ### Author Response · Authors · 2025-11-24
> **We hope that our response addresses your concern**
>
> Dear reviewer QWbo,
>
> We greatly appreciate the time you've invested in reviewing our response. Having submitted our rebuttal, we are eager to know if our response has addressed your concern. As the end of the rebuttal phase is approaching, we look forward to hearing from you for any further clarification that you might require.
>
> Best,
> Submission 9326 authors

---

### Author Response · Authors · 2025-11-30
**Summary of Rebuttals and Revisions for the Area Chair**

Dear Area Chair,

We deeply value the insightful feedback from all reviewers and have taken every comment seriously. For each weakness and question raised by Reviewers, we have provided detailed, direct responses without any evasion. We would be extremely grateful if you could find time amid your busy schedule to review our rebuttal. Below, we summarize how we have resolved the key issues raised by reviewers.

### 1. Robust Generalization & New Baselines (QWbo, v3Mf, Vs1s)
**Concern:** Reviewers questioned the model's ability to generalize beyond the training set (HDTF) and requested comparisons against newer SOTA methods.

**Resolution:**
* **Zero-Shot Cross-Dataset Evaluation:** We conducted new experiments on **VoxCeleb2**, a dataset the model was *never* trained on. Despite this, our method achieved competitive results against SOTA models (Hallo, Hallo2) that include VoxCeleb2 in their training data. This proves out method learns universal motion patterns rather than overfitting.
* **New SOTA Baseline (Hallo2):** We added **Hallo2** suggested by Reviewer v3Mf to our comparison tables. Our method demonstrates comparable or superior performance in image quality (FID/FVD) and identity preservation (CSIM) while maintaining a massive efficiency advantage.
* **Identity Metric:** We added the **CSIM** metric as requested, achieving a score of **0.806**, outperforming DAWN, Hallo, and SadTalker, ensuring high identity preservation.

### 2. Theoretical Grounding & Architecture (Vs1s, PMYf)
**Concern:** Reviewer Vs1s (Soundness: 1) questioned how a Non-Autoregressive (NAR) framework ensures long-term consistency (1600+ frames) without AR mechanisms. Reviewers also sought clarity on the "Microscope" architectural novelty.

**Resolution:**
* **Global Sequence Dependency vs. AR:** We clarified that our NAR consistency is not accidental. The 768$\times$ compression is the critical enabler that allows the *entire* 1600+ frame latent sequence to fit in VRAM. This allows the Diffusion Model (3D-UNet) to model the global mapping from audio to motion latents in parallel. By observing all frames simultaneously, our method mechanistically avoids the **error accumulation** inherent in AR models.
* **The "Microscope" Necessity:** We demonstrated that the two-stage design is not an arbitrary combination but a necessity. We provided empirical evidence (Table 2) showing that single-stage FDE fails at high compression (d/8) due to broken motion assumptions, and single-stage VAE fails (d/32) due to information loss. MDCF is the only configuration that stabilizes this extreme compression.
* **Clarified FDE Mechanics:** We revised Section 3.1 to clarify the FDE's downsampling of the image encoder (regarding the "aggregation" and "compression"), and that the **Occlusion Map ($m$)** is explicitly designed to handle non-warping dynamics (e.g., mouth opening/closing), addressing Reviewer PMYf's concern about flow limitations.

### 3. Validity of Metrics & User Study (v3Mf, Vs1s)
**Concern:** Questions were raised about the fairness of the Flow Smoothness (FS) metric and the scale of the user study.

**Resolution:**
* **FS Metric Validation:** We clarified that FS measures unnatural high-frequency jitter (artifacts) rather than over-smoothing (Figure 5). Our generated videos match the FS profile of Ground Truth videos (0.0803 vs 0.0703) far better than DAWN (0.1211), proving we are restoring natural motion smoothness.
* **Expanded User Study:** We doubled the participant count (20 users) and increased the test video set (10 videos for each method). The results reinforce that our method achieves superior visual stability (V-Stab) and naturalness (O-Nat) compared to baselines.

### 4. Clarification on Hyperparameter Robustness (QWbo)
**Concern:** Reviewer QWbo questioned whether the approach relies on carefully tuned downsampling factors and loss weights, suggesting potential brittleness.

**Resolution:**
* **Loss Weight Stability:** We explained that the loss weights ($\lambda_{KL}=1e^{-4}$, $\lambda_{MSE}=1.0$, $\lambda_{IGC}=1.0$) follow standard training practices rather than extreme fine-tuning.
* **No "Cliff-Edge" Trade-off:** We added an ablation study comparing 192$\times$ compression vs. 768$\times$ compression (Table 2). The results show minimal performance difference, proving that the framework is robust within a reasonable parameter range and does not suffer from a steep trade-off between compression ratio and fidelity.

### Conclusion
We have updated the manuscript to reflect all these changes. We believe the method proposed by our paper now stands as a robust, thoroughly validated framework that solves the critical efficiency bottleneck in high-resolution talking head generation. Finally, regardless of the final outcome of our submission, we sincerely appreciate the time you and all the reviewers have invested, as well as your valuable contributions to the review process.

Sincerely,

Authors of Submission 9326

---

### Meta-Review · Area_Chair_xCHf · 2026-01-04

**Summary:**

The MICROSCOPE framework achieves a major engineering breakthrough, solving the VRAM bottleneck for high-resolution talking head generation. This resource efficiency is the paper's key strength.

However, low initial scores persist due to three fundamental issues:

1. The core flow-based architecture, even with the Occlusion Map, fundamentally limits expressiveness. The visual results still showed similar phenomena of distortion in complex articulations.

2. The justification for NAR coherence across 1600+ frames lacks a formal theoretical guarantee, relying solely on the empirical benefit of compression.

2. The MDCF's structure (flow -> VAE) is conceptually incremental, despite its proven empirical necessity for efficiency.

While the authors satisfied all experimental demands (VoxCeleb2, Hallo2, CSIM), the persistent concerns over visual quality and conceptual novelty restrict any significant score increase.

**Reviewer Concerns:**

Rebuttal successfully closed all major experimental and validation gaps, but conceptual and architectural limitations persist.

> Concerns Addressed:
1. Generalization Risk (QWbo, v3Mf, Vs1s): Resolved.

Added Zero-Shot Cross-Dataset Evaluation on VoxCeleb2, validating universal motion patterns.Missing Baselines/Metrics (v3Mf): Resolved. Added Hallo2 (SOTA baseline) and CSIM (identity metric).

2. Architectural Necessity (PMYf, Vs1s): Resolved.

Ablation proved the MDCF cascade is a necessary engineering invention for the compression (single-stage VAE fails catastrophically).

3. Metric Fairness (v3Mf): Resolved.

Validated Flow Smoothness (FS) metric by matching it to Ground Truth scores, confirming proper measurement of motion jitter.

> Outstanding / Persistent Concerns:

1. Visual Quality/Flow Limitations (PMYf):

The flow-based FDE, even with the Occlusion Map, introduces a ceiling on expressiveness. Visual results also showed similar phenomena of distortion, confirming the artifacts are an inherent limitation.

2. Formal Soundness (Vs1s):

Justification for long-term NAR coherence rests purely on the empirical consequence of compression (allowing parallel processing), lacking a formal theoretical guarantee against error accumulation over 1600+ frames.

3. Conceptual Novelty (Vs1s, PMYf):

MDCF's structure (flow -> VAE) is conceptually incremental, despite its proven necessity for efficiency.

**Reviewer Scores:**

> Reviewer QWbo (4) score will increase.

Empirical concerns (generalization, baselines) fully resolved.

> Reviewer PMYf (4) score no change.

Fundamental flow limitation and persistent visual artifacts remain unaddressed.

> Reviewer v3Mf (4) score will increase.

All specific empirical requirements (data, metrics) flawlessly met.

> Reviewer Vs1s (4) score no change.

Formal soundness and conceptual incrementalism remain unresolved.

---

### Decision · Program_Chairs · 2026-01-26

Reject